



# Does reduction of emissions imply improved air quality?

Cheng Fan [1,3], Zhengqiang Li [1], Ying Li [2], Jiantao Dong[4], Ronald van der A[5,6] and Gerrit de Leeuw [5,1,6,7]

[1]State Environment Protection Key Laboratory of Satellite Remote Sensing, Aerospace Information Research
Institute, Chinese Academy of Sciences, Beijing 100101, China
[2]China Academy of Culture and Tourism, Beijing International Studies University, Beijing 100024, China
[3]University of Chinese Academy of Sciences, Beijing 100049, China
[4]School of Surveying and Land Information Engineering, Henan Polytechnic University
[5]Royal Netherlands Meteorological Institute (KNMI), R&D Satellite Observations, 3731GA De Bilt, The
Netherlands
[6]Nanjing University of Information Science & Technology (NUIST), School of Atmospheric Physics, No.219,
Ningliu Road, Nanjing, Jiangsu, China
[7]University of Mining and Technology (CUMT), School of Environment Science and Spatial Informatics,
Xuzhou, Jiangsu 221116, China

*Correspondence to:* Ying Li (liying20190063@bisu.edu.cn)

**Abstract.** The air pollution in China is among the highest in the world. However, legislation to reduce pollutant
emissions have been successful and concentrations of $SO_2$ (since 2007), aerosols (since 2011) and $NO_2$ (since
2012) have decreased substantially as deduced from satellite and ground-based observations. The strong
reduction of the emissions by the end of January 2020, first caused by the Spring Festival holidays and enhanced
and extended by the COVID-19 lockdown, offered an opportunity to study the effects on air quality across the
country. In particular the reduction of $NO_2$ concentrations observed using the TROPOspheric Monitoring
Instrument (TROPOMI) on board the Copernicus Sentinel-5 Precursor satellite was used to quantitatively
estimate the lockdown effect. To determine a baseline for the expected concentrations in 2020, we used Ozone
Monitoring Instrument (OMI) $NO_2$ TVCD time series from 2011 to 2019 and determined trends for different
regions, noticing a possible halt of the decrease in recent years, but for different periods in the south than in the
north of east China. Neglecting this leveling leads to overestimation of the lockdown effect on the concentrations
in the south, neglecting the trends may lead to underestimation in the north. We also looked at the temporal
resolution used in studies on the reduction of the concentrations and noticed the gradual decrease ahead of the
Spring Festival, which actually continued during 3 weeks into the lockdown. Using satellite observations of
other species and ground-based monitoring data, it was noticed that the expected improvement of air quality due
to the reduction of $NO_2$ concentrations was offset by the increase of the concentrations of aerosols and $O_3$
ascribed to meteorological influences and complex chemistry. In the current study we use TROPOMI
tropospheric $NO_2$ vertical column densities (TVCDs) together with ground-based monitoring data for $NO_2$, $SO_2$,
CO, $O_3$, $PM_{2.5}$ and $PM_{10}$ in 11 regions around large cities to evaluate the evolution of their concentrations during
19 weeks after the Spring Festival and their effect on air quality. For comparison, and in an attempt to average
out short term (e.g., meteorological) influences, ground based monitoring data were used for the same period in
2015-2019. The results confirm that the strong reduction of $NO_2$ does not always imply good air quality because
concentrations of other pollutants may increase. The study shows the different behavior in city clusters in the
north and south of China, and inland in the Sichuan and Guanzhong basins. Effect of other holidays and events
are small, except in Beijing where the air quality in 2020 was notably better than in previous years. This study
was undertaken for east China, but the methodology can also be used for other areas and part of the conclusions
are generally applicable.



## 1. Introduction

Concentrations of aerosols and trace gases in the atmosphere over East China have been increasing in response
to industrial development and urbanization and are among the highest worldwide. However, during the last
decade, air pollution control strategies were implemented as part of a series of government plans to reduce
emissions and thus the concentrations of pollutants (Jin et al., 2016; van der A et al., 2017; Zheng, 2018). As a
result, the concentrations of $SO_2$, $NO_2$ and aerosols have been decreasing during the last decade as concluded
from the analysis of satellite observations (Krotkov et al., 2016; Koukouli et al., 2016; van der A et al., 2017; de
Leeuw et al., 2018; Sogacheva et al., 2018). The nationwide lockdown in response to the COVID-19 outbreak in
January 2020 resulted in a sudden decrease of anthropogenic emissions quantified in Ding et al.(2020). The
unprecedented reduction of the concentrations of $NO_2$, as observed by the TROPOspheric Monitoring Instrument
(TROPOMI) on board the Copernicus Sentinel-5 Precursor satellite, was reported in the media (e.g., NASA
Earth Observatory, 2020; The New York Time, 2020; ESA, 2020; Journalistic platform TU Delft, 2020) and the
scientific literature (e.g., Fan et al., 2020a; Liu et al., 2020; Bauwens et al., 2020). $NO_2$ is an important trace gas
for atmospheric chemistry and a precursor gas for the production of $O_3$ and the secondary formation of aerosols
(Le et al., 2020). Therefore the sudden reduction of the $NO_2$ concentrations together with the concentrations of
other trace gases and aerosols offers a unique opportunity to study the effects of stringent reduction of emissions
on air quality. In particular, the air quality improvement expected from the reduction of the emissions appeared
to be offset by the unexpected increase of aerosol concentrations in the north of China, evidenced by satellite
observations of the aerosol optical depth, a measure for the aerosol loading in the atmosphere (e.g., Fan et al.,
2020a; Huang et al., 2020; Le et al., 2020; Diamond and Wood, 2020). The AOD increase was explained in
terms of meteorological influences, economic impacts and complex chemistry. East China offers a natural
laboratory to study the effects of emission reduction because of its large geographical and climatological
diversity with the North China Plane (NCP), the Yangtze River delta (YRD) and the Pearl River Delta (PRD), as
well as mountain areas with large basins such as the Chongqing/Sichuan and the Guanzhong Basins, all with
high pollution levels, population density and level of industrialization, but different climatological and
meteorological influences on air quality.

The outbreak of the Corona virus SARS-CoV-2, causing a disease named COVID-19, in the winter of 2019/2020,
resulted in a pandemic with strong consequences all around the world. The unknown effect of the virus on health,
its virality spreading and the lack of medicine or vaccine resulted in unprecedented measures to contain the virus
which strongly affected social life and economics. Schools, non-essential businesses and factories were closed,
traffic and public transport were restricted, air traffic was limited and in many countries people were confined to
their homes except for essential trips.

The virus was first discovered in China in December 2019 and reported to the World Health Organization on 31
December 2019 (Bao and Zhang, 2020; Word Health Organization, 2020a). The first strong spreading event was
the outbreak in Wuhan in January 2020 when many people were infected and admitted to hospitals. This led to
the lock-down of Wuhan (23rd Jan, 2020) and additional prefecture cities in Hubei (24th Jan, 2020) and further
restrictions all over China (Bao and Zhang, 2020). Not long after, the virus traveled to other continents resulting
in the lockdown of many countries. The World Health Organization (WHO) declared the pandemic on March
11th 2020 (Word Health Organization, 2020b).





One side effect of the virus containment measures was the reduction of anthropogenic emissions resulting in smaller concentrations of pollutants as observed from space. In particular, reports on the strong reduction of tropospheric $NO_2$ vertical column densities (TVCDs) appeared for China (e.g., Fan et al., 2020a; Liu et al., 2020;

Pei et al., 2020; Zhang et al., 2020a), Italy and other countries in Europe (e.g., Bauwens et al., 2020; Tobías et al., 2020; Mesas-Carrascosa et al., 2020; Sicard et al., 2020; Filippini et al., 2020; Menut et al., 2020) and later over other continents (e.g., Bauwens et al., 2020; Chen et al., 2020; Dantas et al., 2020; Kanniah et al., 2020; Patel et al., 2020; Sharma et al., 2020; Mahato et al., 2020). Reductions in $NO_2$ TVCDs on the order of 50% were reported in the scientific literature (Fan et al., 2020a; Liu et al., 2020; Sicard et al., 2020). Also the effect of the

lock-down on the emission reductions (Ding et al., 2020; Zhang et al., 2020b) and effects on concentrations of other pollutants were reported in the scientific literature (e.g., Fan et al., 2020a; Le et al., 2020), including both satellite and ground-based observations.

In the current study we focus on east China, where the COVID-19 outbreak happened just before the 2020 Spring Festival, or Lunar New Year, China's largest annual holiday which in 2020 fell on 25 January. During the

Spring Festival holidays, which in 2020 started on 24 January and were extended until 10 February, many people return to their families, non-essential industrial activities are stopped or reduced during a week or longer, and educational institutes are closed. As reported in Fan et al. (2020a) this yearly event resulted in 2020 (and the preceding three years) in a decrease of the $NO_2$ TVCDs in east China by about 50% with extremes of up to 70-80% in some areas. Due to the progressive lock-down of China starting just before the Spring Festival (see (Bao and

Zhang, 2020), for time line of events in Hubei and the Jing-Jin-Ji metropolitan circle and its surrounding areas), many people could not return to their workplaces and non-essential businesses remained closed for a longer period, resulting in the augmented reduction of $NO_2$ TVCDs by another 50-70% over east China (roughly east of the HU line) and less over less-populated and industrialized areas in the north and west of China (Fan et al., 2020a). Fan et al. also looked at other species measured by satellites as well as from air quality monitoring

stations in 26 provincial capitals. The satellite measurements showed the reduction of $SO_2$ but the signal was too noisy to deduce a clear quantitative effect. The lockdown did not have a clear effect on the CO TVCDs, except in the south of China. In contrast to $NO_2$, the aerosol optical depth (AOD) increased over the North China Plain (NCP), the most polluted part of China. The ground-based observations, which in the study of Fan et al. are daily averages over all monitoring stations in each of 26 provincial capitals, confirmed the strong reduction of $NO_2$

concentrations by 30-60%, depending on location, with the smallest reduction in the north and west. As expected from chemical considerations, the reduction in $NO_2$ concentrations led to the enhancement of $O_3$ concentrations, which again varied with location, but not in the same way as for $NO_2$. The $SO_2$ concentrations followed those of $NO_2$ but reductions after the lockdown were not as strong and not easy to quantify with a clear pattern. The ground-based CO data confirm the conclusion from the satellite data that there was little effect of the lockdown

other than the usual Spring Festival effect. The same applies to $PM_{2.5}$ for which no substantial reduction was observed due to the lockdown with respect to the reduction usually observed during the Spring Festival.

When the COVID-19 was under control, i.e. the number of new infections decreased, the lock-down measures were gradually relaxed and normal life was resumed step by step. With the increase of mobility and starting up of industry, also the emissions and thus air pollution increased. Several studies reported that air quality was

"back to normal" after 40 days (Bauwens et al., 2020; Filonchyk et al., 2020; Wang and Su, 2020). In the current study we address the question what is "normal", using satellite observations over the last decade over selected





regions, extending to 19 weeks after the 2020 Spring Festival. In addition to satellite data, we use ground-based observations from the Chinese air quality monitoring network providing detailed information in different regions, and compare those for 2020 with similar observations in the last 5 years (2015-2019). The reason for this study

is the gradual decrease of $NO_2$ and $SO_2$ TVCDs (van der A et al., 2017) and AOD (de Leeuw et al., 2018; Sogacheva et al., 2018) during extended periods in the last decades, in response to policy measures by the Chinese Government to reduce emissions and improve air quality. In the estimates of the lock-down effects on air pollution such trends were accounted for by comparison of 2020 with the previous year or years. However, the $NO_2$ TVCDs in early 2020, before the Spring Festival, were much lower than those in 2019 and the question

arises whether the trends derived in earlier studies were continued in more recent years, and whether interannual variations had a stronger effect. In other words, how well can the expected baseline concentrations, serving as reference to determine the reductions due to smaller emissions during the lock down period, be determined?

Another question is whether air quality (AQ) was really improved, in spite of the enormous reduction of $NO_2$ as observed by satellites and confirmed by ground-based monitoring networks. Like for $NO_2$, also $SO_2$ VCDs and

surface concentrations dropped during the Spring Festival. However, the $SO_2$ surface concentrations were rather low before the 2020 Spring Festival and thereafter they were well into the plume of concentrations during the previous 3 years. In contrast, CO VCDs initially remained similar to those before the Spring Festival and surface concentrations decreased during the Spring Festival but did not remain low as $NO_2$ concentrations did. In contrast, the AOD increased, as reported in several studies (Fan et al., 2020a; Huang et al., 2020; Nichol et al.,

2020; Le et al., 2020), which was explained by meteorological influences and/or complicated aerosol chemistry. In contrast to AOD, the $PM_{2.5}$ and $PM_{10}$ concentrations decreased during the Spring Festival and remained low during the lockdown (Fan et al., 2020), suggesting that the processes resulting in the increase of AOD did not have the same effect on PM. Surface $O_3$ concentrations increased (Fan et al., 2020a), in response to the reduced $NO_2$ concentrations shifting the oxidizing capacity. Taking all these differences into account, the question arises

what the effect of the lockdown was on the air quality, as expressed by the air quality index (AQI, see Appendix A for definition) and how AQ or AQI reacted to the gradual release of the socio-economic restrictions.

The objectives of the current study are thus (1) to extend the time series from previous studies to evaluate whether earlier trends can be used to determine baseline concentrations; (2) to determine whether the air quality was indeed improved as much as anticipated from the reduction of tropospheric $NO_2$ TVCDs deduced from

satellite observations; (3) to evaluate whether the pollutant concentrations had returned to normal levels during the study period of 19 weeks after the COVID-19 outbreak, the subsequent lock-down and gradual relaxation of the measures. (4) during these 19 weeks, two significant events occurred in China: the Tomb Sweeping Festival (4-6 April) and the May holidays (1-5 May). In addition the Party Congress took place in Beijing (21-28 May). How did these event influence the air quality?

These objectives are addressed by studying satellite measurements of $NO_2$ TVCDs and monitoring data of $PM_{2.5}$, $PM_{10}$, $NO_2$, $SO_2$, $O_3$, CO and the air quality index (AQI). Time series of monthly averaged $NO_2$ TVCDs for the period 2011-2020 are used, and weekly averages in 2020. Ground-based data are daily averages. The study focuses on 11 regions in east China, mainly around provincial capitals, selected based on the $NO_2$ TVCD levels after about 3 months. However, it is noted that the methodology and part of the ensuing results have generally

applicability and do not only apply over China.





## 2. Methods

### 2.1 Study area

The study area is east China, one of the most polluted regions in the world, for which the air quality was much improved during the COVID-19 lockdown. To monitor the rebound of the concentrations when the lock-down
measures were gradually released, maps were used of weekly-averages of $NO_2$ TVCDs derived from TROPOMI (Figure 3) and their differences with respect to week 0, i.e. the Spring Festival week from 25 to 31 January, 2020 (week numbers are listed in Table A1, difference maps are presented in Figure A1). The difference map for week 12 is shown in Figure 1. The yellow background in this map indicates no changes with respect to week 0, red an increase and green a decrease of the $NO_2$ TVCD. The rectangles are the 11 areas selected for the current study
because of the occurrence of a strong increase or decrease. The regions are listed in Table 1 and include well-known centers such as the Beijing-Tianjin area, Shijiazhuang in west Hebei and Jinan in Shandong, all in the North China Plain (NCP), Shanghai in the Yangtze River Delta (YRD), Guangdong in the Pearl River Delta (PRD), Chongqing and Chengdu, and Wuhan. Each region includes a large city for which monitoring data are available for comparison with the satellite data (Fan et al., 2020a). These regions also provide a reasonable
geographical spread across the country and cover different climate zones.

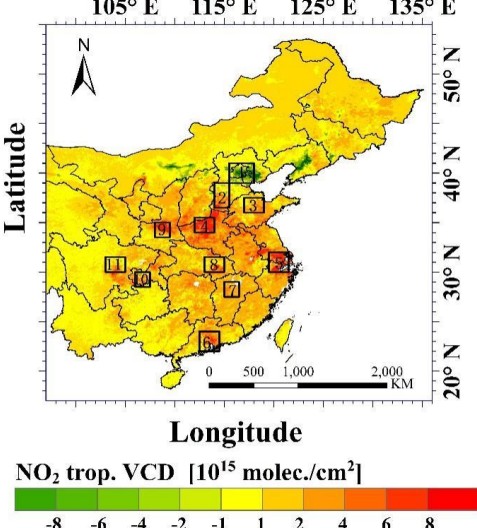

**Figure 1.** Study area showing the $NO_2$ TVCD difference map for week 12 (i.e. $NO_2$ TVCD for week 12 minus $NO_2$ TVCD for week 0). 11 focus regions are indicated with numbers, names and coordinates are provided in Table 1.

**Table 1.** Focus regions of the current study, see Figure 1 for locations corresponding to the numbers in the first column. Coordinates in columns 3 and 4 are the left upper corner of each region, the size around the corner of each region is indicated in columns 5 and 6. Regions are indicated with the name of the central city.

| Nr | Name | Longitude(°) | Latitude(°) | Δ Lon(°) | Δ Lat(°) |
|---|---|---|---|---|---|
| 1 | Beijing-Tianjin | 108.0 | 35.0 | 1.5 | 1.5 |
| 2 | Shijiazhuang | 114.0 | 39.0 | 1.5 | 2.5 |





| 3 | Jinan | 117.0 | 37.5 | 2.0 | 1.5 |
| 4 | Zhengzhou | 112.0 | 35.5 | 2.0 | 1.5 |
| 5 | Shanghai | 119.5 | 32.0 | 2.0 | 2.0 |
| 6 | Guangzhou | 112.5 | 24.0 | 2.0 | 2.0 |
| 7 | Nanchang | 115.0 | 29.0 | 1.5 | 1.5 |
| 8 | Wuhan | 113.0 | 31.5 | 2.0 | 1.5 |
| 9 | Xi'an | 108.0 | 35.0 | 1.5 | 1.5 |
| 10 | Chongqing | 106.0 | 30.0 | 1.5 | 1.5 |
| 11 | Chengdu | 103.0 | 31.5 | 2.0 | 1.5 |

**2.2. Satellite data**

**2.2.1 OMI**

The Ozone Monitoring Instrument (OMI) onboard NASA's Aura satellite was launched in July 2004 (Schoeberl et al., 2006; Levelt et al., 2018). Aura is in a sun-synchronous polar orbit with an equator-crossing time at 13:30 local time (LT). The OMI instrument employs hyperspectral imaging in a push-broom mode to observe solar radiation backscattered by the Earth's atmosphere and surface at 740 wavelengths over the entire range from 270

to 500 nm with a spectral resolution of about 0.5 nm (https://projects.knmi.nl/omi/research/instrument/index.php, last viewed 30 September 2020). With a 2600 km wide swath, OMI provides daily global coverage in 14 orbits. In this study the OMI Quality Assurance for Essential Climate Variance (QA4ECV) version 1.1 product (doi:10.21944/qa4ecv-no2-omi-v1.1) with a 13 x 24 $km^2$ spatial resolution is used (Boersma et al., 2018). This product was validated by, e.g., Lorente et al. (2017) and Zara et al. (2018). The measurement of $NO_2$ is one of

the explicit objectives of the Aura OMI mission. The monthly mean tropospheric $NO_2$ column density data are derived from satellite observations based on slant column $NO_2$ retrievals with the DOAS technique and the KNMI combined modelling/retrieval/assimilation approach (Boersma et al., 2011). $NO_2$ TVCDs for the years 2011 - 2020 were downloaded from the following website: http://www.temis.nl/airpollution/no2.html (last access: 27 September 2020).

**2.2.2 TROPOMI**

TROPOMI is a passive hyperspectral nadir-viewing imager aboard the Sentinel-5 Precursor satellite (also known as Sentinel-5P) launched on 13 October 2017 (Veefkind et al., 2012). Sentinel-5P is a near-polar orbiting sun-synchronous satellite flying at an altitude of 817 km in an ascending node with an equator crossing time at 13:30 LT and a repeat cycle of 17 days. The swath width is approximately 2600 km, resulting in daily global coverage,

with an along-track resolution of 7 km (Veefkind et al., 2012). TROPOMI products used in this study are L3 off-line (OFFL) version products (see http://www.TROPOMI.eu/data-products/ for more detail), in particular tropospheric $NO_2$ column density data for the period around the 2020 Spring Festivals. The spatial resolution at nadir for most products used in this study is 1 km.

The operational validation results are reported every 3 months at the S5P-MOC-VDAF website (http://mpc-

vdaf.TROPOMI.eu/, last access: 27 September 2020). The TROPOMI/S5P tropospheric $NO_2$ column is operationally validated by the S5P-MPC-VDAF (S5P–Mission Performance Centre – Validation Data Analysis





Facility) using the Pandora NO$_2$ total columns from the Pandonia Global Network (PGN). The comparison shows a negative bias of roughly 30%.

### 2.3 Ground-based data

The ground-based data used in this research were downloaded from http://www.pm25.in/ (last access: 27 September 2020), which is the National Real-time Air Quality Publishing Platform public website for air quality monitoring data maintained by the China National Environmental Monitoring Center (CNEMC) of the Ministry of Ecology and Environment of China (MEE, see http://www.mee.gov.cn/, last access: 27 September 2020, for more detail). This website provides PM$_{2.5}$, PM$_{10}$, SO$_2$, NO$_2$, O$_3$, and CO hourly and 24-hour moving averages for

each site or city. Measurement techniques used at the stations, reliability of the data and quality control were briefly described by (Silver et al., 2018) and (Zhai et al., 2019); see also (MEE(Ministry of Environmental Protection of the People's Republic of China), 2012). The data from these websites are provided by local governments and have been used in several studies related to air pollution, air quality, and other aspects in China (Xue et al., 2020; Fan et al., 2020b; Fan et al., 2020a) (http://www.pm25.in/sharer, last access: 27 September

2020). For the current study, we collected hourly PM$_{2.5}$, PM$_{10}$, SO$_2$, NO$_2$, O$_3$, and CO data for 370 cities, with the main focus on the 11 provincial capitals in east and central China as indicated in Section 2.1 and Figure 1, for up to 19 weeks after the Spring Festival, for the years 2015-2020. In the current study, the data collected at different locations in each city were averaged to get a spatially representative number for the whole city, as daily (24 h) averages.

### 3. Results

### 3.1 Satellite Observations

#### 3.1.1 Evolution of NO$_2$ spatial distributions after the 2020 Spring Festival

NO$_2$ tropospheric vertical column densities (TVCDs) were derived from TROPOMI observations. Monthly mean NO$_2$ TVCDs, averaged over 30 days before and after the 2020 Spring Festival are presented in Figure 2. Figure 2

shows the large difference in the NO$_2$ TVCDs before and after the 2020 Spring Festival, similar to those used in Fan et al. (Fan et al., 2020a) for all China to illustrate and analyze the effect of the COVID-19 containment policy measures. Fan et al. concluded that the use of 30-days averages leads to underestimation of the Spring Festival effect and overestimation of the COVID-19 lockdown effect and that for more reliable estimates shorter periods should be used. Therefore, in the current study, weekly NO$_2$ TVCD maps were produced as shown in

Figure 3. Here week numbers relate to the Spring Festival, on Saturday January 25, 2020, i.e. week 0 is January 25-31, week 1 is February 1-7, etc. (see Table A1 for an overview of week numbers and dates). Weeks -1 to -3 are included as references for the NO$_2$ TVCDs during the period before the Spring Festival. The comparison of the monthly TVCDs in Figure 2 with the weekly TVCDs in Figure 3 (top row, week -3 to -1) clearly illustrates the advantage of using better time resolution to show the advancing decline of the NO$_2$ TVCDs in east China

before the Spring Festival. The first lockdown in Wuhan was on 23 January, toward the end of week -1 and therefore the decline was mainly due to the decreasing economic activity associated with the Spring Festival. The combined effects of the Spring Festival and progressive lockdown in east China (Bao and Zhang, 2020) is visible in weeks 1-3, when the NO$_2$ TVCDs were lowest. The slight recovery in week 2 in the south of the study





area may reflect the progressive nature of the lockdowns in different areas in China, i.e. toward the end of the
Spring Festival holidays when people travelled back to their work places when it was still possible.

The maps in Figure 3, and the difference plots with respect to week 0 in Figure A1, show that overall the $NO_2$ TVCDs remained low over all east China during the first two weeks. Also in week 3 the $NO_2$ TVCDs were low, although some increase occurred over industrialized and populated areas in east China (north of the Yangtze River) and in the Guangzhou area which intensified every week from week 4 until week 8. In week 8 the $NO_2$
TVCDs reached high values and the spatial distributions and concentrations changed little during the next 5 weeks, except in week 10 when the $NO_2$ TVCDs were lower (although not in the YRD and Guangzhou). These reduced concentrations may be a sign of reduced emissions during the Tomb Sweeping Festival on 4-6 April. In week 13 the $NO_2$ concentrations were substantially lower than in the weeks before, and this continued in week 14. These weeks encompass the May Festival holiday (1-5 May), another very large national festival in China
when many people travel home to their families. Emissions and concentrations during that time may be lower than in the weeks before. After week 14 the $NO_2$ TVCDs increased in the southern provinces like Hunan and Guizhou as well as in the east around Shanghai, Jiangsu and Shandong, whereas in the northeast the $NO_2$ TVCDs first decreased, then decreased in week 18. Overall, the spatial patterns during these weeks were similar but the TVCDs changed, likely due to changes in economic activity and meteorological influences, but they did
not reach values similar to those before the Spring Festival. However, this would not be expected as discussed in Section 3.1.2.

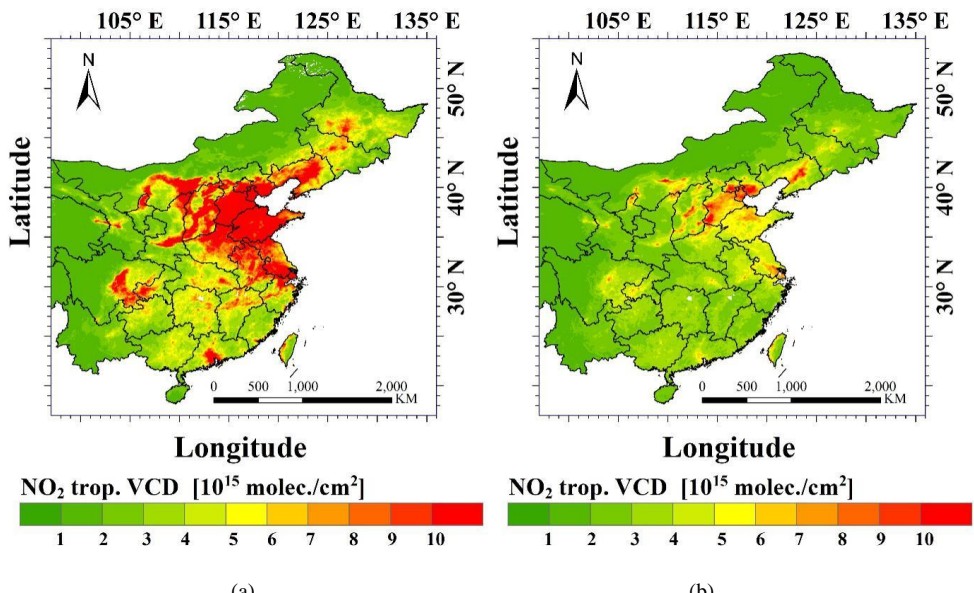

(a)                                                                            (b)

**Figure 2.** NO2 tropospheric vertical column densities over east China averaged over 30 days before (a) and after (b) the 2020
Spring Festival. The 2020 Spring Festival was on 25 January 2020 and thus the period before started on 26 December and the 30-day period after ended on 24 February 2020.







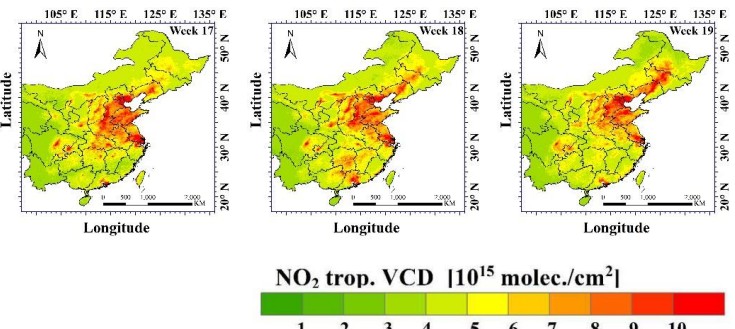

**Figure 3.** Maps of weekly averages of NO2 TVCDs for weeks -3 to -1 and week 0 (top row) and weeks 1-19 in the following rows. Note that week number refers to the 2020 Spring Festival, i.e. week 0 starts on Saturday 25 January 2020 (see also Table S1).

### 3.1.2 NO$_2$ TVCD time series and trends for different regions between 2011 and 2019

#### 3.1.2.1 Monthly mean NO$_2$ TVCD time series

The Spring Festival occurs in the winter when NO$_2$ concentrations reach their highest values. Figure 4 shows time series of monthly mean TVCDs for tropospheric NO$_2$ derived from OMI data for the period from January 1$^{st}$, 2011 until May 31$^{st}$, 2020 for the 11 regions defined in Table 1. For each region, the time series show the strong seasonal variations with sharp peaks in the winter and shallow minima in the summer. The winter TVCD maxima are about a factor of 5 larger than the summer minima, with the ratio varying somewhat by region, with higher values in Shijiazhuang (7.2) and Zhengzhou (6.0), and lowest values of about 2.5 in Guangzhou and Chengdu. These numbers are in reasonable agreement with the factor of 3 reported by Shah et al. (2020) for the NO$_2$ TVCD averaged over central-east China.

On a monthly scale, the TVCD maximum varies a little between regions and years, but in general the peaks occur in December/January and decrease fast toward the summer. For the study of the COVID-19 effects, this implies that, if there would be no restrictions, the NO$_2$ TVCDs would have decreased by a factor of 2-7, depending on region, from the pre-lockdown period to the time when all measures were released. This needs to be taken into account in any study on the effect on air quality during different stages of the COVID-19 lockdown.

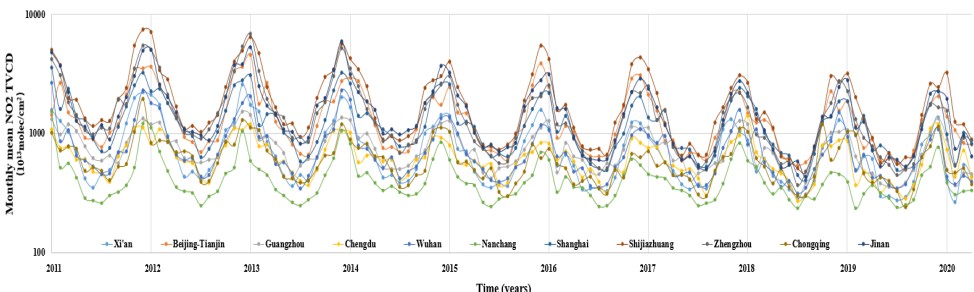

**Figure 4.** Time series of monthly mean NO$_2$ TVCD for 11 regions from January 2011 to May 2020. The NO$_2$ TVCDs are plotted on a logarithmic scale to better visualize the differences between different regions as well as the gradual variation of the TVCDs during the summer months.



### 3.1.2.2 Trends of annually averaged tropospheric $NO_2$ TVCDs

Figure 4 shows that the $NO_2$ TVCDs vary strongly by region, with the highest TVCDs in Shijiazhuang, Zhengzhou and Jinan (in 2012), although the relative differences changed from year to year. The winter-peak TVCDs seem to have decreased between 2012 and 2017, whereas in the years 2017-2019 they were of similar magnitude, i.e. the decrease seems to have come to a halt. Similar behavior is observed in the summer months. However, the time series suggest that the period of decreasing $NO_2$ TVCDs and the occurrence of the maximum and minimum values was not the same for all regions. To further investigate trends in different regions and the differences between them, time series were plotted for each region and, to reduce effects of short term (monthly) variations, this was done for annual mean $NO_2$ TVCDs. The results in Figure 5 show a grouping with very high $NO_2$ TVCDs in the north of the study area and Shanghai, with a clear separation from the lower TVCDs in the other regions as expected from the spatial maps in, e.g., Figure 2a. Another noticeable difference between the regions is the clear distinction between the temporal behavior in the north and in the south. In the regions in the north, i.e. in the NCP (Shijiazhuang, Beijing, Jinan and Zhengzhou) and Xi'an the TVCDs were similar in 2011, 2012 and 2013; from 2013 they decreased exponentially until 2018 (Xi'an until 2016). In the other regions, i.e. Shanghai and those in the south and west, the TVCDs decreased exponentially from 2011 until they reached a minimum value in 2015 or 2016 after which they remained low (e.g., Shanghai, Nanchang) or even increased somewhat (e.g., Chongqing). Overall, after 2016 the TVCDs in these regions fluctuated from year to year but remained within 10% of the values in 2015 (except in Chongqing). In view of these differences, trend lines to the annual mean $NO_2$ TVCD data in regions in the north were fitted for the years 2013-2018, whereas for the other regions trend lines were fitted for the years 2011-2015 or 2016. The results are presented in Table 2, where the trend (year$^{-1}$) describes an exponential decrease of the TVCDs following the relationship $y = a\,e^{bt}$, where $y$ = TVCD, $a$ is the intercept (TVCD in first year of the fitting period), $b$ is the trend (year$^{-1}$) and $t$ is the number of years after year1. Coefficients of determination ($R^2$) are all high and the trend lines in Figure 5 show the good fit. The data in Figure 5 also show that beyond the fit interval the TVCDs do not follow the trend for that region and level off as discussed above.

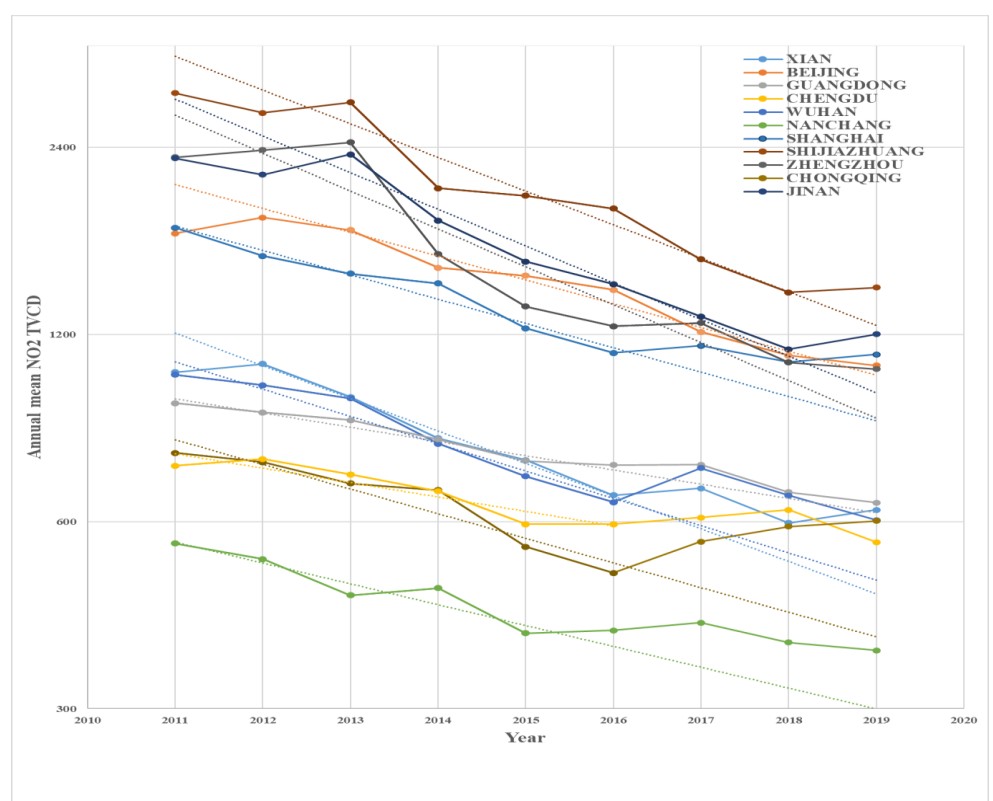

**Figure 5.** Time series of annual mean $NO_2$ TVCDs for each of the 11 regions, plotted on a semi-logarithmic scale and fitted exponential trend lines ($y = a \cdot e^{bt}$) for different periods as described in the text. The periods and trends are listed in Table 2.

**Table 2.** $NO_2$ TVCD trends determined for the period in the 3rd column. Trends are determined using exponential fits ($y = a \cdot e^{bt}$) to time series of the annual mean $NO_2$ TVCDs as shown in Figure 5 and described in the text.

| Number | Region | Period | Trend (year$^{-1}$) | $R^2$ |
|---|---|---|---|---|
| 1 | Beijing-Tianjin | 2013-2018 | -0.090 | 0.96 |
| 2 | Shijiazhuang | 2013-2018 | -0.125 | 0.92 |
| 3 | Jinan | 2013-2018 | -0.136 | 0.96 |
| 4 | Zhengzhou | 2013-2018 | -0.140 | 0.81 |
| 5 | Shanghai | 2011-2016 | -0.090 | 0.97 |
| 6 | Guangzhou | 2011-2015 | -0.050 | 0.95 |
| 7 | Nanchang | 2011-2015 | -0.080 | 0.90 |
| 8 | Wuhan | 2011-2016 | -0.100 | 0.96 |
| 9 | Xi'an | 2012-2016 | -0.120 | 0.99 |
| 10 | Chongqing | 2011-2016 | -0.090 | 0.92 |
| 11 | Chengdu | 2011-2016 | -0.053 | 0.88 |

Having established that the annually averaged TVCDs decrease exponentially during a certain period of time and change little during more recent years (in the southern regions) or the last year (2019, in the northern regions),





we need to determine whether these conclusions also apply to shorter periods of time during which effects of the lock-down on the concentrations of atmospheric trace gases are studied. As a compromise between high time resolution and reducing meteorological effects on concentration differences, monthly averaged $NO_2$ TVCDs were selected and plotted as time series for the 11 study areas. Because of the focus of this study on the recovery of the air quality after the lockdown measures were released, and because the signals in the summer months are relatively weak, this was only done for the winter months. Furthermore, to exclude the effect of the Spring Festival on the $NO_2$ TVCDs, January and February were excluded. This left November, December, March and April for 2011-2019 and the results are presented in Figure A2. The data in Figure A2 show the overall decline of the $NO_2$ TVCDS, following the yearly trends and variations of the annual mean TVCDs in Figure 5 and the differences between the 11 regions for different periods. Overall, the periods when the $NO_2$ TVCDs decrease are similar to those indicated in Table 2, although for Beijing and Shijiazhuang a strong minimum is observed in 2014 which may be associated with emission reduction because of the Asia-Pacific Economic Cooperation (APEC) meeting in Beijing in November 2014 and the China Victory Day Parade in September 2015. Interannual variability is stronger in the monthly mean data than in the annual means, as expected. Because of these variations, trend lines for monthly mean $NO_2$ TVCDs were not computed. The main message is that the TVCDs follow the tendencies in the annual means with leveling toward the end of the study period. The $NO_2$ TVCDs are highest in December of almost all years, followed by November (it is noted that January was not included in these figures because of expected deviations in response to reduced emissions during Spring Festival holidays). During March and April the $NO_2$ TVCDs are approaching the summer minima and although the TVCDs decline between 2012 and 2019, the trends are quite small.

### 3.2 Effects on air quality: ground-based observations

### 3.2.1 Time series of air quality index for 11 regions

The air quality index (AQI) is based on the mass concentrations of $PM_{2.5}$, $PM_{10}$, $NO_2$, $SO_2$, CO and $O_3$ as described in Appendix A. AQI is determined by only one pollutant, i.e. the pollutant with the highest AQI. Time series for the AQI in the 11 cities identified in Table 1 are plotted in Figure 6, for 15 weeks after the Spring Festival in 2020. Tianjin was added as a second megacity in the metropolitan agglomeration because of its potentially different AQ due to large industrial activities as opposed to the capitol city. Also plotted are AQI time series for the same weeks in the years 2015-2019. Together the time series of the five previous years form a plume which serves as reference for the 2020 time series. As shown in Figure 6, there are large variations between the years and there is no specific ordering indicating a systematic temporal variation (tendency). Hence the plume is representative for the range of variations that can be expected in 2020 (provided that 2020 is not an exceptional year in regard of factors that influence the AQ, such as meteorological factors). Figure 6 also shows that the width of the plume varies between different cities. It is noted, that the AQIs are weekly averages over all measurements in each city, created from 24-hour averages at each site.

Comparison of the time series in Figure 6 shows the similarity between the AQI time series in the five cities in the NCP (Beijing, Tianjin, Shijiazhuang, Jinan and Zhengzhou), for the cities Shanghai, Guangzhou and Wuhan (group 2), and for Nanchang, Xi'an, Chongqing and Chengdu (group 3). In view of these similarities, a grouping is made between these cities for further analysis. In the NCP, the behavior of most pollutants contributing to the





AQI in 2020, as well as that with respect to the plumes, is similar. For the other two groups this is not always obvious and some deviations may occur for different pollutants.

In 2020, the AQI in the cities in the NCP group fluctuates in the first 3 weeks and then stays low until week 9;

increases toward the plume in week 10 and then stays at the bottom of the plume. Except in the first 3 weeks, the AQI is smaller than 100 indicating good air quality. For group 2, the AQI fluctuates and indicates excellent to good AQ until week 10 when the AQI index moves into the plume or occasionally above (Guangzhou) but still indicating good AQ. For group 3 the AQI indicates good AQ, except in Xi'an (moderate) and decreases somewhat (Xi'an becomes "good") but remains in the plume throughout the whole study period. In other words,

the AQI does not indicate better AQ for these cities in response to COVID-19 containment measures.

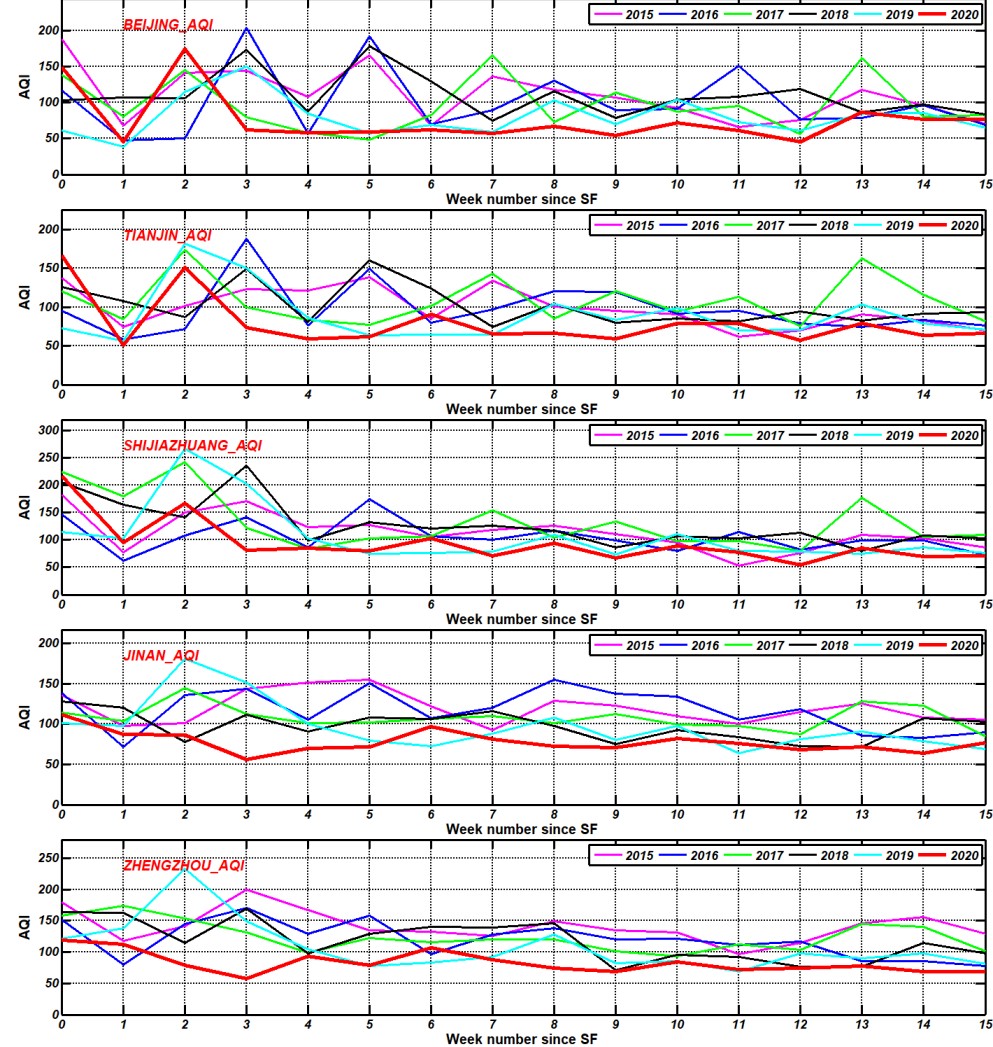













**Figure 6.** Time series of the AQI in 12 cities for the first 16 weeks (week 0 to week 15) after the Spring Festival in 2020, together with AQI time series for the same weeks in 2015-2019. See legend for identification. Note that the vertical scales
vary between different cities.

### 3.2.2 Time series of aerosols and trace gases affecting air quality

With the AQI determined by the pollutant with the highest AQI,  which may not necessarily be the species observed from satellites such as $NO_2$, the behavior of the individual pollutants contributing to the AQI will be considered using time series similar to those in Figure 6 for the AQI. Because of the similarity between the time
series for the cities in each of the three groups, the individual time series will be discussed for one city in each group: Shijiazhuang, Wuhan and Chongqing. These cities are selected after inspection of time series of the 6 individual pollutants; it is noticed that dissimilarities may occur between cities in each group. This detail may be discussed when necessary.

Time series for the weekly averaged concentrations of the 6 pollutants measured in Shijiazhuang, Wuhan and
Chongqing are presented in Figures 7, 8 and 9, respectively. Note that for some pollutants the vertical scales may be different between the three cities. Overall, the concentrations of most pollutants during the five reference years (further referred to as the plume) were higher in Shijiazhuang than in Wuhan and Chongqing. This applies to aerosols ($PM_{2.5}$ and $PM_{10}$), $SO_2$ and CO, but not for $NO_2$ and $O_3$ for which the concentrations in the plumes in these three cities were similar. This is remarkable because the satellite $NO_2$ TVCDs in 2016 were about a factor
of 3 higher in Shijiazhuang than in Wuhan which in turn were about 30% higher than in Chongqing (Figs 4 and 5) whereas in 2019 the $NO_2$ TVCDs were a factor of 2.3 higher in Shijiazhuang than in the other two regions where the TVCDs were similar. However, for an adequate comparison between satellite data and surface concentrations, and thus effects on AQ, factors influencing the relation between near surface concentrations and TVCDs need to be accounted for, such as meteorological factors driving vertical mixing. Also long-term trends,
inter-annual variations, seasonal variations, local emissions and meteorological effects influencing (photo)chemical reactions determining the overall concentrations need to be considered.

In all three cities the surface $NO_2$ concentrations are overall decreasing during the study period following the seasonal variation which is also observed in the satellite data (Figure 4). However, in the satellite data the decrease is largest when concentrations are highest but, as discussed above, for the ground-based data the
concentration differences between the three cities are not large. Yet, the seasonal effect seems more pronounced in Shijiazhuang than in Wuhan than in Chongqing. This is also different from the satellite data (see Figure 4) and may be a life time effect related to lower temperatures in the north than in Wuhan and Chongqing.

Another difference between the three cities is the effect of the lock-down on the surface $NO_2$ concentrations. In Shijiazhuang the plume decreased after week 3 whereas the 2020 concentrations increased steadily from week 0
and the curve joined the plume in week 6 although remaining near the bottom of the plume. In Wuhan the 2020 concentrations were far below the plume (20 $\mu g \cdot m^{-3}$ vs 40-60 $\mu g \cdot m^{-3}$) until week 9 after which they suddenly increased in week 10 to remain just below the plume (~40 $\mu g \cdot m^{-3}$). In Chongqing the $NO_2$ concentrations were low (20 $\mu g \cdot m^{-3}$) during the first 3 weeks, then increased and remained close to the plume in weeks 6 to 9 and merged into the plume from week 10. The temporal behavior of the ground-based $NO_2$ concentrations was
similar to the temporal development in the satellite TVCDs in Figure 3.

When $NO_2$ concentrations decrease, $O_3$ concentrations rise as is observed for all three cities. However, there was no substantial difference between the temporal variation of the $O_3$ concentrations in 2020 and the other years.





The 2020 concentrations were inside the plume during the whole study period and no anomalous behavior was observed in spite of the reduced $NO_2$ concentrations during the first 6-10 weeks. Rather, in Shijiazhuang and

Chongqing both the $NO_2$ and $O_3$ concentrations were low in the plume. The $O_3$ concentrations in Shijiazhuang and Wuhan were similar and a bit higher than in Chongqing.

The $SO_2$ concentrations in the three cities were different by a factor of 2-3. In Shijiazhuang they were highest followed by Wuhan where the $SO_2$ concentrations were similar to those in Chongqing. The common seasonal variation with high concentrations in the winter and low in the summer (Koukouli et al., 2016; Wang et al.,

2018b; Zhang et al., 2019) is reflected by the relatively fast decrease in the first 8-10 weeks after which they remained low at 20-30 $\mu g \cdot m^{-3}$ in Shijiazhuang and around 10 $\mu g \cdot m^{-3}$ in the other two cities. In all three cities there was much interannual variation with substantially lower concentrations in 2018 and 2019 than in the years before. In 2020, the variation of the $SO_2$ concentrations during the study period and the behavior with respect to the plume in the three cities were different. In Shijiazhuang the $SO_2$ concentrations were low (<20 $\mu g \cdot m^{-3}$)

throughout the whole study period, with small variations, and did not merge with the plume. It is noticed that also in 2019 the $SO_2$ concentrations in Shijiazhuang were low. In Chongqing the concentrations during the first 5 weeks ( ~7 $\mu g \cdot m^{-3}$) were below the plume and merged with the plume after week 7. In contrast, in Wuhan there was no evident lockdown effect on the $SO_2$ concentrations, i.e. during the lockdown period they were similar to those in 2019 and 2020 (7-8 $\mu g \cdot m^{-3}$). After week 6 the 2020 concentrations increased somewhat and merged

inside the plume of the previous 4 years.

The plume concentrations of CO during the first weeks of the study period were higher in Shijiazhuang than in Wuhan and Chongqing, whereas toward the end they were lowest in Shijiazhuang (between 0.5 and 1 $mg \cdot m^{-3}$). As opposed to the behavior of the other trace gases, in 2020 the CO concentrations were low in the plume but not clearly affected by the lockdown.

For aerosols the situation was different than for the trace gases. The data in Fig 7 show that in Shijiazhuang $PM_{2.5}$ was relatively high during the first 3-4 weeks during all 5 years, and those in 2020 were well inside the plume. Thereafter the $PM_{2.5}$ concentrations dropped and, apart from some fluctuations, remained low (on average about half of those in the first weeks) and those in 2020 were almost every week near the bottom of the plume. It is noted that the $PM_{2.5}$ concentrations in 2017 and 2019 were substantially higher than in other years.

$PM_{10}$ behaved qualitatively similar to $PM_{2.5}$ but the concentrations in the first weeks were not much higher than later in the study period. In contrast, the 2020 concentrations of both $PM_{2.5}$ and $PM_{10}$ in Wuhan in weeks 2 and 3 were lower than in any other week during the study period and also lower than in all 5 years before (about 1/3 of the plume average). In Chongqing the $PM_{2.5}$ concentrations were well inside the plume (around the average) and the plume decreased gradually as expected from the common seasonal behavior of $PM_{2.5}$. $PM_{10}$ was also inside

the plume although somewhat closer to the bottom. Hence in Chongqing the COVID-19 lockdown measures did not have an evident effect on the aerosol concentrations, in spite of the strong reduction of $NO_2$ concentrations. The $PM_{2.5}$ concentrations in week 0 were 150 $\mu g \ m^{-3}$ in Shijiazhuang, 50 $\mu g \ m^{-3}$ in Wuhan and 40 $\mu g \ m^{-3}$ in Chongqing, for $PM_{10}$ the concentrations were 200, 50 and 50 $\mu g \ m^{-3}$, respectively.




**Figure 7.** Time series of the concentrations of PM2.5, PM10, NO2, SO2, CO and O3 in Shijiazhuang for weeks 0 to 15 starting from the Spring Festival in 2020 (red line), together with time series for these pollutants for the same weeks in 2015-2019. See legend for identification.




**Figure 8.** As Figure 7, but for Wuhan.





**Figure 9.** As Figure 7, but for Chongqing.

## 4. Discussion

The satellite data on tropospheric NO₂ VCDs and ground-based monitoring data for the concentrations of NO₂
and other trace gases, aerosols and AQI all indicate the different behavior of atmospheric composition in the

north and south of China and the selected regions have been grouped to discuss the characteristic behavior within
each group. This relates both to long-term variation (trends in the satellite data) and the reaction to the COVID-



19 lockdown. Hence the answers to the questions we set out with for this study at the end of the introduction will be different for each of the regional clusters which emerged in the course of the study.

### 4.1 Estimation of lockdown effects: effects of temporal resolution

Many studies on the COVID-19 lockdown effect on atmospheric concentrations are based on comparison of a period before and after the start of the lockdown or on comparison with the same period in previous years. The lockdown occurred during the Spring Festival holidays during which the concentrations of $NO_2$, often used in studies on the effect of the COVID-19 lockdown, are typically reduced by 50%. Hence, in many studies the Spring Festival effect was separated from the total effect to determine the effect of the lockdown only. One way

to do this was presented in Fan et al. (2020) for tropospheric $NO_2$ VCDs and total column $SO_2$ and CO VCDs, while in contrast AOD was observed to increase. The AOD increase was anticipated to be due to meteorological factors conducive to the formation of haze. Also the ratio of the $PM_{2.5}$ concentrations before the Spring Festival to those after were higher than in previous years (in Beijing $PM_{2.5}$ even increased by a factor of 2.2) and were higher in north China than in the south. In their estimates on the lockdown effects from satellite data, Fan et al.

(and others) used averages over the months before and after the Spring Festival. The current study clearly shows that during the weeks before lockdown the $NO_2$ TVCDs gradually decreased and varied also after the lockdown. The post Spring Festival variations are also observed in the surface concentrations of $NO_2$ which varied by more than a factor of 2. This also applies to aerosols and $SO_2$ concentrations, and to a lesser extend to CO and $O_3$. Hence the actual effect of the lockdown on the concentrations of aerosols and trace gases will be influenced by

the separation from the Spring Festival effect, the temporal resolution chosen for the data analysis, as well as the correction for meteorological and other factors such as reduction of emissions and related concentration trends as well as chemistry. The influence of emissions and the impact of the shutdown on different economic sectors on $NO_2$ and aerosol concentrations was discussed by Diamond et al. (2020).

### 4.2 What is the baseline for the estimation of lockdown effects?

Decadal time series of monthly and annual mean $NO_2$ TVCDs for the 11 regions and the annual trends derived from these were presented in Section 3.1.2 (Figures 5 and S2, Table 2). For the calculation of the baseline concentrations in 2020, i.e. the concentrations expected if there would not have been a lockdown, the seasonal variation needs to be taken into account. However, monthly trendlines are difficult to determine with some accuracy due to interannual variations and due to the Spring Festival effect which occurs at different dates in the

solar calendar. Therefore trends for January and February were nor considered. Furthermore, as Figure 5 shows, the decline in the $NO_2$ TVCDs seems to level off in recent years, i.e. after 2015 in the south of China and possibly after 2018 in the north. The years when the trends were changing are similar for the monthly and annual mean data. Hence the baseline could be determined using an average over the years after the trend change. The uncertainty in these averages is about 10% (Figure 5). Ignoring the trend change, i.e. assuming that the trend

would continue to 2020, would result in an underestimation of the baseline for 2020. Extrapolation of the trend for Wuhan to 2020 would result in an estimated baseline of $440 \cdot 10^{13}$ molec·$cm^{-2}$ and for Chongqing $357 \cdot 10^{13}$ molec·$cm^{-2}$, whereas using the average over 2016-2019 for 2020, i.e. assuming that the decrease has halted as suggested by the data in Figure 5, would provide a baseline of $660 \cdot 10^{13}$ molec·$cm^{-2}$ for Wuhan and $561 \cdot 10^{13}$ molec·$cm^{-2}$ for Chongqing. In other words, ignoring the trend change would result in a baseline lower by about





35% and thus in an overestimation of the lockdown effect on the $NO_2$ TVCD. Similar considerations may apply to Shijiazhuang, but considering that a change in the annual trend did not occur until 2018, the variation in following years is difficult to estimate. In view of this discussion, the use of a climatology over recent years for comparison with the 2020 concentrations may be a good strategy for regions in the south of China, whereas for the north, where concentrations were decreasing until 2018, the climatological concentrations may be too high.

The use of ground-based data leads to larger uncertainties. As the ground-based data in Figures 8 and 9 for Wuhan and Chongqing show, the $NO_2$ concentrations in the plumes vary strongly from week to week and the plume width is therefore rather large, with an uncertainty which is much larger than the 10% uncertainty in the trend since 2015/2016.

Meteorological influences may be twofold. Meteorological conditions may be conducive of the formation of

haze in stagnant air as often observed in north China during the winter (e.g., Li et al., 2018; Wang et al., 2019; Wang et al., 2018a; Wang et al., 2020b). On the other hand, large scale weather systems influence the transport of air masses from different origins transporting clean air or pollution contributing to local air quality (e.g., Wang et al., 2019; Li et al., 2018). Another aspect is the influence of air temperature, humidity and radiation on chemistry which affects $NO_2$, $O_3$ and aerosols, in particular for the situation during the COVID-19 lockdown

with the strong reduction of $NO_2$ concentrations. A reduction of $NO_2$ (or $NO_x$ = NO + $NO_2$, where NO is only a small fraction of $NO_x$) leads to an increase in $O_3$, as observed in the ground-based data. The enhanced $O_3$ concentrations result in the increase of the oxidizing capacity of the atmosphere which in turn leads to the production of secondary organic aerosol (SOA) as explained in, e.g., Diamond et al. (2020) and Le et al. (2020). The increased aerosol concentrations result in the attenuation of solar irradiation due to more scattering and

absorption which in turn may further influence the meteorology (Zhong et al., 2018) and photochemical reactions.

In view of the decisive role of meteorology in haze formation in north China (Le et al., 2020) it is surprising that both Le et al. (2020) and Diamond et al. (2020) used meteorological data averaged over 1 month (February 2020). Haze occurs episodically and less than 25% of the episodes last longer than 4 days (Wang et al., 2018c;

Wang et al., 2020a).

**4.3 Lockdown effect on air quality and rebound**

The similarity in the temporal behavior of the air quality index (Figure 6) was the basis for the subdivision of the regions in three groups. In the NCP group the AQI fluctuated in the first 3 weeks and reached a peak value in week 2, then remained low. The peak was highest (170, moderately polluted) in Beijing where it exceeded the

value of previous years. Obviously, this was due to a haze episode with strongly enhanced $PM_{2.5}$ with respect to the period before the Spring Festival (Fan et al., 2020a) and a concentration of ca. 140 μg·m$^{-3}$, almost double the 24-h class 2 (for cities) air quality standard in China specified in GB 3095-2012 (https://www.transportpolicy.net/standard/china-air-quality-standards/, last access 27 September 2020). In other cities in the NCP the peak values were lower, decreased with distance to Beijing, and were also lower than in

2017 and 2019. Only in Jinan and Zhengzhou the $PM_{2.5}$ values were within the 24-h class 2 (for cities) air quality standard. After week 3 the $PM_{2.5}$ concentrations were within air quality standard limits and the AQI was between 50 and 100 (good) and lower than in the previous years, for all cities in the NCP. However, closer inspection shows that the $O_3$ concentrations exceeded the air quality standard of 100 μg·m$^{-3}$ (1-hour mean value) between



weeks 5 (in the south of the NCP) and 7 (in the north). Furthermore, in all cities in the NCP the $O_3$
concentrations in 2020 were well inside the plume. Hence, in the NCP the strong emission reduction during the
lockdown and the strong decrease of $NO_2$ concentrations observed both from space and from the surface
monitoring network, were offset by the increase of other pollutants. Early in the lockdown the aerosol
concentrations were high due to meteorological conditions and complex chemical influences, and later the $O_3$
concentrations exceeded the limiting values. However, the latter were not reflected in the AQI, which followed
the variations in $PM_{2.5}$ but remained low when $O_3$ concentrations were high. In fact, for all cities in the NCP the
AQI was below or just inside the plume during the whole study period, whereas the $NO_2$ concentrations moved
into the plume toward the end, except in Beijing. The relatively low $NO_2$ concentrations might be expected based
on both the decreasing trends in the $NO_2$ TVCDs until 2018 in the north of China and from the seasonal decrease.
Also $SO_2$ concentrations remained low, close to those in 2019. With these considerations, it is hard to determine
whether the pollutant concentrations in the NCP returned to their normal levels, which in regard of seasonal
variations are expected to be lower than before the lockdown and in regard of their decreasing trends are
expected to be lower than in other years or, considering that the trends level off, similar to those in the last
couple of years.

For the Group 2 cities, Shanghai, Guangzhou and Wuhan, the AQI during the lockdown varied and AQ was
good until week 10, with the largest effect in Wuhan. $NO_2$ concentrations in Wuhan were very low with 20
$\mu g \cdot m^{-3}$ during the first 9 weeks (3 times lower than the plume average). In Shanghai and Guangzhou the
concentrations were initially similar but increased slowly. In all three cities the $NO_2$ concentrations merged into
the plumes after week 10, more or less coincident with the end of the lockdown after 76 days, on April 8, 2020.
$PM_{2.5}$ was not reduced as much as $NO_2$ but was also below the plume and overall traced the $NO_2$ concentrations,
moving into the plume after week 10. Being further south than the NCP, $O_3$ concentrations were close to the air
quality standard of 100 $\mu g \cdot m^{-3}$ and exceeded that limit around week 5, as in the NCP. Hence, also in the Group 2
cities the reduction of other pollutant concentrations was offset by the increase of $O_3$ which is not reflected in the
AQI. The rebound at the end of the lockdown period is clear with all indicators returning to levels similar to
those in the earlier years, i.e. inside or close to the plume.

Group 3 includes three cities in the Sichuan/Chongqing and Guanzhong Basins and Nanchang. In these cities the
AQI was not substantially affected by the lockdown, except in the very beginning when it was low inside (or
even below) the plume but overall remained inside the plume. Yet, the $NO_2$ concentrations were around 20 $\mu g \cdot m^{-3}$
during the first 3 weeks, initially some 40-60% lower than the plume for which the concentrations actually
increased during these 3 weeks and then gradually decreased. Between week 3 and 7 the $NO_2$ concentrations in
2020 increased in all 4 cities to about 50 $\mu g \cdot m^{-3}$ in week 7, close to the plume, and later merged into the plume.
$SO_2$ concentrations were close to those in 2017. $PM_{2.5}$ was not much different from the plume throughout the
whole period and in all three cities. Decreasing somewhat in the basins and fluctuating around 40 $\mu g \cdot m^{-3}$ in
Nanchang. $O_3$ concentrations were lower than 100 $\mu g \cdot m^{-3}$ (50 $\mu g \cdot m^{-3}$ in Chongqing) and gradually increased to
above 100 $\mu g \cdot m^{-3}$ around week 7. Overall, the lockdown had little effect on the air quality in Group 3 in spite of
the significant reduction of the $NO_2$ concentrations. The latter returned to normal levels after about 9 weeks.
The differences between the lockdown effects on the air quality in the three clusters has not been analyzed in
detail. The duration of the lockdown was not exactly the same in each city. For instance, in Xi'an peoples' lives



gradually returned to normal during a period of 1 month ending on 27 March (Zhang et al., 2020a) whereas in Wuhan the lockdown ended on 8 April. The effects of the gradual increase of activities depends on the kind of
activity and resulting emissions. The effect of the emissions on the concentrations depends on meteorological conditions and other factors influencing dispersion of the pollutants such as the local topography in the basin area which limits transport.

### 4.4 Effects of national holidays and other events

The 15 weeks study period covered the lockdown from the beginning (week 0) to the end of the lockdown in
Wuhan on 8 April (week 10) and the last weeks were included to monitor the rebound of the pollutant concentrations and, for Beijing and Tianjin, to include the Party Congress which took place in Beijing, 21-28 May, 2020 (i.e. during weeks 16-17). During the 15 week period, also two national holidays occurred, the Tomb Sweeping Festival (4-6 April, i.e. in week 10) and the May holidays (1-5 May, i.e. during weeks 13-14). The Tomb Sweeping Festival was just before the end of the lockdown in Wuhan when $NO_2$ concentrations were
observed to rise from very low to close to the values observed in previous years (Figure 8) and also concentrations of other pollutants as well as AQI peaked, as discussed in Section 3.2. As shown in Figure 6, in most other cities the AQI was a little higher in week 10 but the effect was not strong. Also during the May holidays in weeks 13-14 the AQI and the concentrations of other pollutants do not stand out. Although during these holidays families usually get together there was no significant effect on AQ, possibly because the
concentrations were already lower and had not fully recovered, while on the other hand travel in 2020 was still restricted.

The situation in Beijing was different. With the regular occurrence of large (inter)national meetings, emission control measures are often enforced in Beijing and, except during the haze event in the first weeks of the lockdown, the AQI was low (around 50), well below the plume, until week 13 (Figure 6). In weeks 13 and 14 the
AQI merged into the plume as did the concentrations of all pollutants, except $NO_2$ which however also increased in that period. The subsequent decrease of the concentrations resulted in minima in the concentrations in week 17. This is illustrated in Figure 10 showing time series of the AQI and ground-based concentrations of $NO_2$ and $PM_{2.5}$ in Beijing from week 0 to week 19.

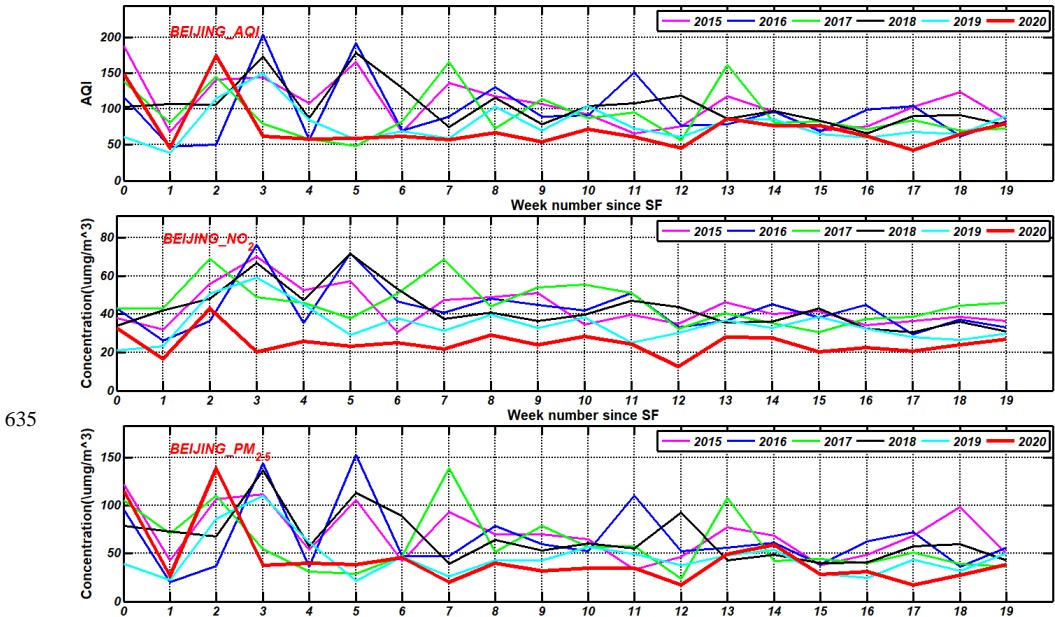

**Figure 10.** Time series of AQI and the concentrations of NO2 and PM2.5 in Beijing, from week 0 to week 19.

### 5. Conclusions

China is a fast developing country with a high degree of urbanization and industrialization, especially in east China. China also offers a large variety of meteorological, climatic and geographical conditions with fast plains, large mountainous terrain and desert areas. The country can roughly be divided by the Yangtze River, with different influences on the atmospheric processes in the north than in the south. These effects came out in the analysis of both the satellite data and the ground-based monitoring data of trace gases and aerosols affecting air quality. In addition, the Sichuan and Guanzhong basins have their own characteristics due to the influence of the surrounding mountains. Because of the similar evolution of the AQI and pollutant concentrations, the cities were grouped in three groups: the NCP group, Group 2 in the south, and Group 3 including two major basin areas. The focus of the current study was the evolution of the concentrations of the pollutants during the last decade in response to emission reduction policies, and in particular the effect of the sudden reduction in economic activities and peoples' mobility during the lockdown in response to the COVID-19 outbreak in China. Clearly, a very large effect was observed on the concentrations of $NO_2$, both near the surface and tropospheric column densities, whereas the concentrations of aerosols and $O_3$ were observed to increase. Hence the question came up whether the air quality was really improved as much as suggested from the $NO_2$ data, or that this reduction was offset by the increase of the concentrations of other pollutants? To answer these questions, satellite and ground-based data were analyzed for 11 regions in east China, leading to the following conclusions.

1. The effect of the lockdown is often determined by comparison of the concentrations before and after the lockdown, averaged over a certain period of time which often is taken as 1 month to average out short-term effects. However, as shown in Figure 3, the concentrations evolve over much shorter time scales and decrease from week to week, both before and after lock down. A complicating factor for the



determination of the lockdown effect was that the lockdown happened during the Spring Festival holidays during which the concentrations already decreased substantially. The lockdown added more severe reductions during a much extended period of time. Hence, for a proper determination of the evolution of the concentrations a temporal resolution of 1 week is more suitable than 1 month. The choice of shorter time scales is even more important when episodical events occur, such as the haze event early in the lockdown in the north of China, and for the use of meteorological data.

2. Time series of annual mean tropospheric $NO_2$ VCDs for 2011-2019 for the 11 regions show the decreasing trend in the north between 2014 and 2018, and in the south between 2011 and 2015/2016. After these periods, the decrease was halted and concentrations fluctuated but remained below the 2015 values. Trends vary between -0.05 and -0.14 per year (see Table 2).

3. To determine the effect of the lockdown on the concentrations, a baseline must be determined which depends on both long term trends and short term variations. Ignoring the trend change in the south would result in an underestimation of the baseline concentrations of the order of 35% as compared to using the average of the concentrations in 2016-2019 as baseline. In the north the baseline is difficult to determine because the $NO_2$ TVCDs continued to decrease until 2018. Using the average of the concentrations in 2016-2019 as baseline would result in an overestimate of the baseline concentrations.

4. The effect of the strong reduction of the $NO_2$ concentrations on the air quality is offset by the increase of $O_3$ concentrations and aerosols ($PM_{2.5}$). The increase of aerosols in the north of China has two reasons: the meteorological conditions conducive to the formation of haze and the complicated chemistry involving $NO_2$ and $O_3$, leading to the formation of secondary aerosols.

5. The effect of the lockdown is different in cities in the NCP Group, Group 2 and Group 3, with an increase AQI in the first weeks in the NCP group and improved AQ later. The concentrations did not return to the levels in the previous four years (the plume), but it is hard to determine which baseline to expect. In Group 2 the AQ was substantially improved during about 10 weeks, although after week 5 the effect of the reduced concentrations was offset by the increase of $O_3$ exceeding National Ambient Air Quality Standards. After the lockdown the concentrations returned to levels similar to those in the plume. For Group 3 cities the concentrations were initially reduced but after a few weeks increased to inside the plume. Hence, apart from the first weeks, the lockdown did not have a significant effect on the AQ in the Group 3 cities, in spite of the substantial reduction of the $NO_2$ concentrations which returned to normal levels after about 9 weeks. The use of AQI is questionable because its definition is not following a physical quantity: even when AQI indicates good AQ, limits may be exceeded.

6. Holidays like the Tomb Sweeping Festival and the May holidays are expected to have some effect on the air quality, but in 2020 this was hardly noticeable. However, in Beijing the air quality during the Party Congress, at the end of May, was better than during the weeks before. It is noted that throughout the whole study period of 19 weeks, the $NO_2$ concentrations in Beijing were strongly reduced with respect to those in the preceding 4 years.

This study was undertaken for east China, but the methodology and results can in part also be applied to other areas. In particular this applies to the temporal resolution, which in this study was taken as 1 week, as opposed to 1 day or 1 month in earlier studies such as Fan et al. (2020a). As discussed above, meteorological variations influencing air quality, such as formation and dissipation of haze, take place on rather short time scales. Whereas,





the determination of effects of sudden changes in emissions on pollutant concentrations needs to consider such
short term meteorological effects as well as interannual changes, the baseline correction using multi-annual data
also needs to account for (changes in) long term trends and seasonal variations as discussed in detail for $NO_2$.

**Appendix A.**

The air quality index (AQI) is based on measured mass concentrations of aerosols and trace gases. The method
to determine the AQI is described in (Yuan and Yang, 2019) and is calculated according to the National
Standards on Air Quality Measurement published by the Chinese Ministry of Environmental Protection on 29
February 2012 -Ambient air quality standards (GB 3095-2012) (MEE(Ministry of Environmental Protection of
the People's Republic of China), 2012) and the Technical Regulation on Ambient Air Quality Index (on trial)
(HJ 633-2012) (MEP(Ministry of Environmental Protection of the People's Republic of China), 2012) that
became effective on January 1st, 2016). The aerosol mass concentrations considered are $PM_{2.5}$ and $PM_{10}$ and the
trace gases are $NO_2$, $SO_2$, $O_3$ and CO. The AQI is calculated using the method described in (MEP(Ministry of
Environmental Protection of the People's Republic of China), 2012). The individual AQI of each of these 6
pollutants ($IAQI_P$) is calculated using (Yuan and Yang, 2019):

$$\text{IAQI}_P = \frac{IAQI_{Hi} - IAQI_{Lo}}{BP_{Hi} - BP_{Lo}} (C_P - BP_{Lo}) + IAQI_{Lo}, \tag{A1}$$

Where $C_P$ is the mass concentration of pollutant P, $BP_{Hi}$ and $BP_{Lo}$ are the higher and lower threshold of pollutant
concentration near $C_P$ corresponding to specified IAQI (Individual Air Quality Index) regulated by government
policy. $IAQI_{Hi}$ and $IAQI_{Lo}$ are the corresponding IAQI to $BP_{Hi}$ and $BP_{Lo}$, respectively. The AQI is the highest of
the 6 individual $IAQI_P$:

$$\text{AQI} = \max \{IAQI_1, IAQI_2, \dots , IAQI_6\} \tag{A2}$$

Eq (A2) shows that the AQI reflects only one pollutant, with the highest IAQI, and is not a combination of all 6
(Yuan and Yang, 2019).

An AQI of 50 means that the air quality is excellent, and AQI between 50 and 100 means it is good. When 100<
AQI< 150 the AQ is lightly polluted, for 150<AQI<200 AQ is moderately polluted and for 200<AQI<300 AQ is
heavily polluted. AQI > 300 indicates severe pollution (MEP(Ministry of Environmental Protection of the
People's Republic of China), 2012).

**Table A1.** Definition of the weeks around the Chinese Spring Festival, as used in this study, from 2015 to 2020. The first day
of week 0 in each year is defined as the first day of the Lunar New Year, which in the solar calendar changes from year to
year.

|        | 2015 | 2016 | 2017 | 2018 | 2019 | 2020 |
|--------|------|------|------|------|------|------|
| Week-3 | 01.29-02.04 | 01.18-01.24 | 01.07-01.13 | 01.26-02.01 | 01.15-01.21 | 01.04-01.10 |
| Week-2 | 02.05-02.11 | 01.25-01.31 | 01.14-01.20 | 02.02-02.08 | 01.22-01.28 | 01.11-01.17 |
| Week-1 | 02.12-02.18 | 02.01-02.07 | 01.21-01.27 | 02.09-02.15 | 01.29-02.04 | 01.18-01.24 |
| Week 0 | 02.19-02.25 | 02.08-02.14 | 01.28-02.03 | 02.16-02.22 | 02.05-02.11 | 01.25-01.31 |
| Week 1 | 02.26-03.04 | 02.15-02.21 | 02.04-02.10 | 02.23-03.01 | 02.12-02.18 | 02.01-02.07 |
| Week 2 | 03.05-03.11 | 02.22-02.28 | 02.11-02.17 | 03.02-03.08 | 02.19-02.25 | 02.08-02.14 |
| Week 3 | 03.12-03.18 | 02.29-03.06 | 02.18-02.24 | 03.09-03.15 | 02.26-03.04 | 02.15-02.21 |
| Week 4 | 03.19-03.25 | 03.07-03.13 | 02.25-03.03 | 03.16-03.22 | 03.05-03.11 | 02.22-02.28 |
| Week 5 | 03.26-04.01 | 03.14-03.20 | 03.04-03.10 | 03.23-03.29 | 03.12-03.18 | 02.29-03.06 |





| | | | | | |
|---|---|---|---|---|---|
| Week 6 | 04.02-04.08 | 03.21-03.27 | 03.11-03.17 | 03.30-04.05 | 03.19-03.25 | 03.07-03.13 |
| Week 7 | 04.09-04.15 | 03.28-04.03 | 03.18-03.24 | 04.06-04.12 | 03.26-04.01 | 03.14-03.20 |
| Week 8 | 04.16-04.22 | 04.04-04.10 | 03.25-03.31 | 04.13-04.19 | 04.02-04.08 | 03.21-03.27 |
| Week 9 | 04.23-04.29 | 04.11-04.17 | 04.01-04.07 | 04.20-04.26 | 04.09-04.15 | 03.28-04.03 |
| Week 10 | 04.30-05.06 | 04.18-04.24 | 04.08-04.14 | 04.27-05.03 | 04.16-04.22 | 04.04-04.10 |
| Week 11 | 05.07-05.13 | 04.25-05.01 | 04.15-04.21 | 05.04-05.10 | 04.23-04.29 | 04.11-04.17 |
| Week 12 | 05.14-05.20 | 05.02-05.08 | 04.22-04.28 | 05.11-05.17 | 04.30-05.06 | 04.18-04.24 |
| Week 13 | 05.21-05.27 | 05.09-05.15 | 04.29-05.05 | 05.18-05.24 | 05.07-05.13 | 04.25-05.01 |
| Week 14 | 05.28-06.03 | 05.16-05.22 | 05.06-05.12 | 05.25-05.31 | 05.14-05.20 | 05.02-05.08 |
| Week 15 | 06.04-06.10 | 05.23-05.29 | 05.13-05.19 | 06.01-06.07 | 05.21-05.27 | 05.09-05.15 |
| Week 16 | 06.11-06.17 | 05.30-06.05 | 05.20-05.26 | 06.08-06.14 | 05.28-06.03 | 05.16-05.22 |
| Week 17 | 06.18-06.24 | 06.06-06.12 | 05.27-06.02 | 06.15-06.21 | 06.04-06.10 | 05.23-05.29 |
| Week 18 | 06.25-07.01 | 06.13-06.19 | 06.03-06.09 | 06.22-06.28 | 06.11-06.17 | 05.30-06.05 |
| Week 19 | 07.02-07.08 | 06.20-06.26 | 06.10-06.16 | 06.29-07.05 | 06.18-06.24 | 06.06-06.12 |


**Figure A1.** Difference plots for weekly averages of NO₂ TVCDs minus that for week 0 (Figure 3) for weeks 1-19. Note that week number refers to the 2020 Spring Festival, i.e. week 0 starts on Saturday 25 January 2020.





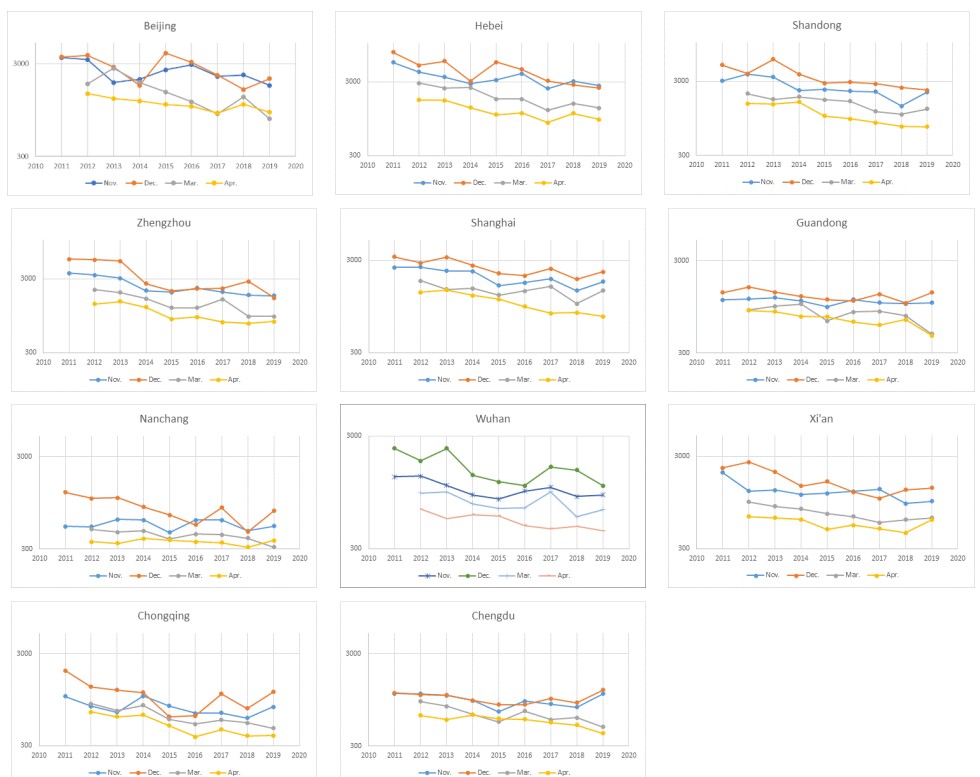

**Figure A2.** Time series of monthly averaged NO$_2$ TVCDs over each of the 11 regions, for each of the months November, December, March and April from 2011 to 2019.

### Data availability statement

QA4ECV data are available at https://doi.org/10.21944/qa4ecv-no2-omi-v1.1 .

The OMI data are available via https://www.temis.nl.

The TROPOMI data are available via http://www.TROPOMI.eu/data-products/.

The ground-based monitoring data are available via http://www.mee.gov.cn/.

### Author contributions

CF, YL and GdL designed the study which was conducted by CF and YL. JTD contributed to the data presentation. CF, YL and GdL prepared the manuscript, ZQL and RvdA provided extensive comments and suggestions for the manuscript. All authors discussed the results and read and commented on the paper.

### Competing interests

The authors declare that they have no conflict of interest.

### Acknowledgements

The authors thank the TEMIS team for maintaining the website and the free use of tropospheric NO$_2$ column data from the OMI sensor from www.temis.nl and the NASA Earth Science Division for the free use of



tropospheric products from the TROPOMI (Sentinel-5P) sensor. We also acknowledge the China National
Environmental Monitoring Center (CNEMC) of the Ministry of Ecology and Environment of China (MEE) for
the provision of the ground-based monitoring data at http://www.mee.gov.cn/. We thank Abdelrazek Elnashar of
Cairo University for the Google Earth Engine technical support. The authors thank anonymous reviewers for
their effort to critically review the manuscript and providing constructive comments.

**Financial support**

This work was supported by the National Outstanding Youth Foundation of China (Grant No. 41925019) and the
National Natural Science Foundation of China (Grant No. 41671367).

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
