# Peer review of "Variability of NO2 concentrations over China and effect on air quality derived from satellite and ground-based observations"

_Atmospheric Chemistry and Physics, 2020_

## Referee Comment (RC1) · Anonymous Referee #1 · 29 Dec 2020

I have reviewed the manuscript "Does reduction of emissions imply improved air quality". In this work, authors used TROPOMI tropospheric NO2 together with ground-based monitoring data in 11 regions around large cities to evaluate the evolution of their concentrations during 19 weeks after the Spring Festival and their effect on air quality. Based on the review criteria of ACP, my comments are as follow. General comments: 1, the content of this article hardly supports the title. The title of this paper has an implicit meaning that a significant reduction in emissions occurred during the closure period. Generally speaking, it is right. However, the difference in the degree of reduction of emissions of different species directly determines their impact on the atmospheric environment. This work only provides evidence on the reduction of NOX, which has already been reported in many previous works. Reductions or variations

of other pollutants are necessary to evaluate Changes in emission sources. 2, This work posed more questions than it answered. Actually, in February and March 2020, many researchers in China, even including many average Chinese, had already intuitively noticed that the air quality does not seem to have improved significantly after the pandemic lockdown, and there are even signs that it is getting worse. After June, such a feeling was confirmed by many research papers. Now, people are now curious as to why this phenomenon has occurred. I would encourage authors to answer this question but not to "discover" it again.

Specific comments: 1, the introduction is lengthy and lacks the necessary logic, which makes it a bit hard to follow. Besides, Lines 69-81 are not relevant with this work. Maybe, authors can supplement a table in the Appendix or section 2 to describe lockdown measures and period in different regions. 2, the study area is called as "east China" as shown in figure 1. I think the co-called "east China" covers parts of North, Northeast, Central, East, South, Southwest and even Northwest China. When I saw the term "east China", I thought this work focused on Yangtze River Delta. 3, regarding extraction of baseline of NO2, authors should utilized some professional analytic tools rather than simply averaging. See "Long-term trend and variability of atmospheric PM 10 concentration in the Po Valley". 4, similar with comment 3, it is necessary to carry out hypothesis tests to draw conclusions on whether air quality returned to the normal level. Simply showing the change curve does not lead to any statistically significant conclusions. 5, the quality of figure 5 is too low to be published in an academic journal. 6. This work hardly provided evidences regarding changes in meteorological conditions. As far as I know, meteorological conditions play a role in determining air quality that is nearly as significant as that of emissions.

---

## Referee Comment (RC2) · Anonymous Referee #2 · 2 Jan 2021

The introduction section is too long and moves from one topic to the next, mixing them at points in a confusing manner; there is a discussion on the current state of the air quality over China, a discussion on other COVID-19 related works, a discussion on aerosols, SO2 and CO, mixed with a discussion on how the measures were enforced, all of them stating names of cities and provinces with which most readers are not familiar with. In lines 82 to 92, references are given to other works as big chunks when, most of these works do not report similar findings, nor are they based on similar methodologies. Then exhaustive details are given from the Fan et al., 2020a, paper which again mention other species and confuse the issue, in a rather long paragraph. As a result, I am already perplexed as to what this article is adding to our current knowledge on air quality effects in China due to COVID-19. In the text, SO2 is greatly

discussed and I fail to see why the author's didn't also analyse SO2 as part of this study, in line with the manuscript title which refers to air quality in general. Two different satellite sensors' tropospheric NO2 VCDs were used as data input for the manuscript's hypothesis. It is well known that the two sensors have a bias in their findings, not only due to the different spectral/spatial characteristics, but also due to the algorithm. How was this bias accounted for? Even for cases of the same algorithm being used on different satellite records, a careful homogenisation is needed to be able to discuss trends in a meaningful manner, see for e.g. the work of Georgoulias et al., 2019, ACP - Trends and trend reversal detection in 2 decades of tropospheric NO2 satellite observations (copernicus.org). The section on satellite data is hence mismatched. The TROPOMI are shown as maps of periods in 2020, the OMI data as timeseries analysis over different regions for a different time period, not including 2020, without the two actually coming together to a coherent conclusion. The manuscript then provides a long discussion on air quality over different Chinese locations based on the AQI calculated from ground-based stations. This section does not merge in the least with the rest of the text, nor with the abstract and the satellite section. Overall the feeling is that this manuscript started as an AQI-based paper with the satellite analysis added on top without the two merging. The discussion section reads out of sequence as well, since the reader is forced to go back and forth to figures presented in previous sections. Many parts are also non-sequiturs where they should appear first in the text, such as the meteorology effects. The continuous references to the work of Fan et al., 2020a, leads the reader to think that this manuscript aspires to be a Part 2 of that work. I suggest to the co-author team to consider what the main take away message should be and re-write/re-present their work accordingly.
* * *

---

## Author Comment (AC1) · 1 Feb 2021

I have reviewed the manuscript "Does reduction of emissions imply improved air quality". In this work, authors used TROPOMI tropospheric NO2 together with ground- based monitoring data in 11 regions around large cities to evaluate the evolution of their concentrations during 19 weeks after the Spring Festival and their effect on air quality. Based on the review criteria of ACP, my comments are as follow.

*R: We thank the reviewer for the time spent on thoroughly reading the manuscript and providing constructive comments. Below we repeat the comments and we structured them in separate paragraphs to systematically provide a response to each of them (in italics and* `red`*).*

*Because of the many changes, including both new text and deleting text and figures, the inclusion of all changed parts would require to append almost the whole revised text. Therefore, we provide line numbers referring to the clean version of the revised text, which will be uploaded when the editor invites us to do so. Shorter texts are copied below.*

**General comments**:

1, the content of this article hardly supports the title. The title of this paper has an implicit meaning that a significant reduction in emissions occurred during the closure period. Generally speaking, it is right. However, the difference in the degree of reduction of emissions of different species directly determines their impact on the atmospheric environment. This work only provides evidence on the reduction of NOX, which has already been reported in many previous works. Reductions or variations of other pollutants are necessary to evaluate Changes in emission sources.

*R: We agree with the reviewer that the title did not fully cover the content. Although several publications point at the reduction of emissions, we did not determine emissions, we only used concentrations (ground-based), or tropospheric column densities (from satellites OMI and TROPOMI). We disagree that we only showed reduction of NO2, we also showed other species in the ground-based data. However, we do feel that indeed not all species show a clear effect on the air quality (or AQI). Therefore we removed SO2, CO and PM10 (including figures) and only focused on NO2, O3 and aerosols (PM2.5) which are interconnected through chemical processes as explained in the revised introduction, while also the effects of socio-economic changes during the lockdown were different for NO2 and PM.*

*In response, we have **changed the title** to "Variability of $NO_2$ concentrations over China and effect on air quality derived from satellite and ground-based observations" and **rewritten most of the Introduction** (Lines 42-127) with a better focus on the actual work presented in the manuscript, and removed most of the general descriptions of the lockdown effects. We also **changed and shortened the abstract** (Lines 18-41), and made **quite some changes throughout the whole text**, and Sect. 5 **Conclusion** was changed to Sect. 5 **Summary and conclusion** (Lines 581-657). We **moved the many AQI time series plots to the Appendix** (Figure A3).*

2, This work posed more questions than it answered. Actually, in February and March 2020, many researchers in China, even including many average Chinese, had already intuitively noticed that the air quality does not seem to have improved significantly after the pandemic lockdown, and there are even signs that it is getting worse. After June, such a feeling was confirmed by many research papers. Now, people are now curious as to why this phenomenon has occurred. I would encourage authors to answer this question but not to "discover" it again.

*R: We agree that many papers have been published on the lockdown effects of the concentrations of pollutant concentrations in China and indeed some of them specifically focused on the reduction of air quality. We have referenced the papers that we were aware of (probably not all of them and we may have missed some, we have added some more recent papers in the revised version). We also note that these papers discussing the AQ getting worse, mainly dealt with the air quality in the NCP: different reasons for the reduction of air quality, in spite of the enormous reduction of $NO_2$ concentrations (or TVCDs), were proposed. These were mentioned in the initial version of our manuscript, but probably not explicitly enough. We certainly did not want to claim that we "discovered" this phenomenon. In the revised version we discuss this subject more extensively in the introduction (first paragraph), together with references to published papers (Lines 54-69).*

*Furthermore, we notice also (Introduction) the connection between the reduction of $NO_2$ concentrations and the increase of $O_3$ and the secondary formation of aerosols, which is the reason to focus in the revised version on these three species. We realized that the abundance of information on other species was distracting from the main work, and therefore removed most of that. (Lines 54-69; 109-114).*

*We also have explained more explicitly the reasons for extending our study (Fan et al., 2020) but we removed much of the detail. The shorter version can be found in lines 76-94. At the end of the introduction we summarized these reasons in four clear objectives (Lines 115-122)) and refer to these in the conclusions (Lines 607-648).*

*In summary, we do not "discover" again the lack of improvement of air quality, but present the data in detail and discuss the effects of pollutants on AQ (as expressed by AQI), i.e. whether there was indeed improved AQ and if it was improved, how long did it last?*

**Specific comments:**

1, the introduction is lengthy and lacks the necessary logic, which makes it a bit hard to follow. Besides, Lines 69-81 are not relevant with this work. Maybe, authors can supplement a table in the Appendix or section 2 to describe lock- down measures and period in different regions.

*R: As mentioned in our response to the general comments, the introduction has been rewritten extensively with a focus on the actual study presented in this manuscript. Less relevant text has been removed including summaries of the spreading of the COVID virus and the lockdown measures. In fact we did not really use this detail in the study and so we decided that it is not useful to provide it. The detail can be found in other publications and we cited some which are most relevant.*

2, the study area is called as "east China" as shown in figure 1. I think the co-called "east China" covers parts of North, Northeast, Central, East, South, Southwest and even Northwest China. When I saw the term "east China", I thought this work focused on Yangtze River Delta.

*R: In the submitted version, east China was defined as the part of China East of the Hu line. This was a bit hidden in the text and probably the reviewer did not notice this. Therefore, we now define the study area explicitly and have also drawn the Hu line in Figure 1:* "In the current study we focus on the part of mainland China east of the HU line (Figure 1), further referred to in this paper as east China, where 94% of the Chinese population lives (Chen et al., 2016)." (lines 130-131, Figure 1)

3, regarding extraction of baseline of $NO_2$, authors should utilized some professional analytic tools rather than simply averaging. See "Long-term trend and variability of atmospheric PM 10 concentration in the Po Valley".

*R: We are aware of this publication by Bigi et al. (2014). However, we used methods which are commonly used by others for the purpose of determining trend lines (e.g. Sogacheva et al., 2018, https://doi.org/10.5194/acp-18-16631-2018). We note that the current study is not intended to provide accurate trends but rather to indicate the effect of the use of earlier data on the determination on a baseline value.*

4, similar with comment 3, it is necessary to carry out hypothesis tests to draw conclusions on whether air quality returned to the normal level. Simply showing the change curve does not lead to any statistically significant conclusions.

*R: The "normal level" differs for each country and region. Here we do not refer to AQ standards but to the AQ before the lockdown, where we use the levels in 5 previous years (2015-2019) as reference. And likewise for the concentrations of species contributing to AQ. This is mentioned in Sect. 3.2.1 (Lines 34-336):* "AQI time series for the same weeks in the five previous years (2015-2019) were plotted to form a plume which serves as reference for the 2020 time

series.*") and 3.2.2 (Lines 362-364: "*AQI time series for the same weeks in the five previous years (2015-2019) were plotted to form a plume which serves as reference for the 2020 time series.*").

5, the quality of figure 5 is too low to be published in an academic journal.

*R: We apologize for the bad quality of the original Figure 5. We have re-drawn Figure 5 and added some explanation in the caption. We hope that the reviewer is satisfied with the current version. In addition, we also improved Fig A2, in the same style as Fig. 5*

6. This work hardly provided evidences regarding changes in meteorological conditions. As far as I know, meteorological conditions play a role in determining air quality that is nearly as significant as that of emissions.

*R: We have mentioned the effect of meteorological conditions many times throughout the whole manuscript. We have not explicitly taken meteorological effects into account. This requires the use of a transport model which is not in the scope of this study. Instead, we have used averages to reduce effects of meteorology on our results, such as the 5-year plumes in the ground-based time series (see response to comment 4) or the annual averages in the decadal time series for $NO_2$ TVCDs (Lines 282-284:* "To further investigate trends in different regions and the differences between them, time series were plotted for each region and, to reduce effects of short term (monthly) variations, this was done for annual mean $NO_2$ TVCDs.").* Furthermore, a brief discussion was included on meteorological influences. To give this more emphasis, the title of Sect. 4.2 has been changed to* "4.2 Long term trends, trend reversal and meteorological influences on the estimation of lockdown effects".* Meteorological influences are discussed in lines 4982-498.*
* * *
*We thank the reviewer for the time spent on thoroughly reading the manuscript and providing critical comments. Below we repeat the comments and we structured them in separate paragraphs to systematically provide a response to each of them (in italics and red).*

*Because of the many changes, including both new text and deleting text and figures, the inclusion of all changed parts would require to append almost the whole revised text. Therefore, we provide lines numbers referring to the clean version of the revised text, which will be uploaded when the editor invites us to do so. Shorter texts are copied below.*

*We do agree with many comments and in response have substantially edited the manuscript, whereas we do not agree with some other comments. However, these comments may have been made because the initial version of our manuscript was not clear enough. With the revised version, in response to both reviewers, the text more clearly explains our reasons for the research undertaken, resulting in 4 objectives, how these were addressed, and what we concluded from this work in relation to the initial research questions.*

The introduction section is too long and moves from one topic to the next, mixing them at points in a confusing manner; there is a discussion on the current state of the air quality over China, a discussion on other COVID-19 related works, a discussion on aerosols, SO2 and CO, mixed with a discussion on how the measures were enforced, all of them stating names of cities and provinces with which most readers are not familiar with. In lines 82 to 92, references are given to other works as big chunks when, most of these works do not report similar findings, nor are they based on similar methodologies. Then exhaustive details are given from the Fan et al., 2020a, paper which again mention other species and confuse the issue, in a rather long paragraph. As a result, I am already perplexed as to what this article is adding to our current knowledge on air quality effects in China due to COVID-19.

*R: Thank you for this comment. We agree that the Introduction was too long and not to the point. We have **revised most of the introduction and removed much text** to provide a more clear and logical storyline to introduce our work (Lines 42-127). We also **changed the title** to better reflect the contents of the manuscript: "Variability of $NO_2$ concentrations over China and effect on air*

*quality derived from satellite and ground-based observations". The manuscript now focuses mostly on NO2 variations on decadal (2011-2019) and weekly (the lockdown period and thereafter) scales. In addition we added O3 and PM2.5, for reasons explained in the text (variation, effect on AQ) (Lines 54-69; 109-114) and removed SO2, CO and PM10 which do not add information or do not show much variability (CO and SO2) (Lines 54-69; 109-114). We also **changed and shortened the abstract** (Lines 18-41), and made **quite some changes throughout the whole text**, and the **conclusion** was changed to **summary and conclusion** (Lines 581-657). We **moved the many AQI time series plots to the Appendix** (Figure A3).*

*We realize that readers may not be familiar with names and locations of regions or cities in China. However, a map is provided in Figures 1 and names are listed in Table 1:* "The names of the regions shown in Figure 1 and their geographical locations are listed in Table 1 and include well-known centers such as …" (Sect. 2.1, Lines 140-143). *In the revised version we have made frequent reference to this information.*

In the text, SO2 is greatly discussed and I fail to see why the author's didn't also analyse SO2 as part of this study, in line with the manuscript title which refers to air quality in general.

*R: SO2 figures and text have been removed as indicated in the response to the previous comment, and as discussed in more detail in the revised manuscript (Lines 88-93).*

Two different satellite sensors' tropospheric NO2 VCDs were used as data input for the manuscript's hypothesis. It is well known that the two sensors have a bias in their findings, not only due to the different spectral/spatial characteristics, but also due to the algorithm. How was this bias accounted for? Even for cases of the same algorithm being used on different satellite records, a careful homogenisation is needed to be able to discuss trends in a meaningful manner, see for e.g. the work of Georgoulias et al., 2019, ACP - Trends and trend reversal detection in 2 decades of tropospheric NO2 satellite observations (copernicus.org). The section on satellite data is hence mismatched. The TROPOMI are shown as maps of periods in 2020, the OMI data as timeseries analysis over different regions for a different time period, not including 2020, without the two actually coming together to a coherent conclusion.

*R: We agree that if we would use the two instruments in the same analysis we should first homogenize these data sets. Especially since these two instruments have another resolution, a homogenization as in Georgoulias et al. (2019) would be needed. However, in our study the measurements of the two instruments are not combined. TROPOMI data is used to analyze 2020 data only, to study the effects of the COVID-19 regulations in section 3.1.1. In section 3.1.2 only OMI data is used to derive trends for the period 2011-2019. This information is added in the (new) pre-amble to Sect. 2.2:* "Two satellite products were used in this study, i.e. the tropospheric NO₂ vertical column densities (NO₂ TVCDs) from OMI and TROPOMI. These products are briefly discussed in the following sub-sections. The OMI NO₂ TVCDs were used for time series analysis over the period 2011-2019, the TROPOMI NO₂ TVCDs, with better spatial resolution, were used to visualize weekly averaged spatial variations of the NO₂ TVCDs and discuss their evolution over the study area. OMI and TROPOMI products thus provide complementary information for different periods of time and were used for different purposes." *To avoid any misunderstanding, the source of the data (OMI or TROPOMI) is now indicated in the captions of Figures 2-5.*

The manuscript then provides a long discussion on air quality over different Chinese locations based on the AQI calculated from ground-based stations. This section does not merge in the least with the rest of the text, nor with the abstract and the satellite section. Overall the feeling is that this manuscript started as an AQI-based paper with the satellite analysis added on top without the two merging.

*R: Thank you for this comment. In fact, the study started out with the satellite work, in particular the TROPOMI data which formed the basis for the further continuation of the study, with questions on the temporal resolution (months, weeks) and the actual starting point which provides the base line for the evaluation of the lockdown effect and how to separate this from the reduction of the initial reduction of the NO2 concentrations during the Spring Festival holidays. We then added the ground-based data for more detail on the effects on AQ.*

*Probably this was not clearly presented in the original manuscript and in therefore, in the revised manuscript we have described this more clearly (Lines 97-101:* "In the current study we address the question what is "normal", using satellite observations over the last decade over selected regions, extending to 16-20 weeks after the 2020 Spring Festival. In addition to satellite data, we use ground-based observations from the Chinese air quality monitoring network providing detailed information in different regions, and compare those for 2020 with similar observations in the last 5 years (2015-2019).").

The discussion section reads out of sequence as well, since the reader is forced to go back and

forth to figures presented in previous sections. Many parts are also non-sequiturs where they should appear first in the text, such as the meteorology effects. The continuous references to the work of Fan et al., 2020a, leads the reader to think that this manuscript aspires to be a Part 2 of that work.

*R: We do not quite understand this comment. It is common to first present the data and explain them in the "Results" section, after which they are discussed in the "Discussion" section. This follows the instruction on the ACP website (https://www.atmospheric-chemistry-and-physics.net/submission.html#manuscriptcomposition), point 5: "**Sections**: the headings of all sections, including introduction, results, discussions or summary must be numbered".*

*Meteorological effects are not a main part of the work but need to be accounted for. Therefore, we have mentioned the effect of meteorological conditions many times throughout the whole manuscript. We have not explicitly taken meteorological effects into account. This requires the use of a transport model which is not in the scope of this study. Instead, we have used averages over the 5 previous years as reference, i.e. the 5-year plumes in the ground-based time series, where the plume includes effects of meteorology, or the annual averages in the decadal time series for $NO_2$ TVCDs. This is mentioned in Sect. 3.2.1 (Lines 34-336):* "AQI time series for the same weeks in the five previous years (2015-2019) were plotted to form a plume which serves as reference for the 2020 time series.") *and 3.2.2 (Lines 362-364:* "AQI time series for the same weeks in the five previous years (2015-2019) were plotted to form a plume which serves as reference for the 2020 time series."). *Furthermore, a brief discussion was included on meteorological influences. To give this more emphasis, the title of Sect. 4.2 has been changed to* "4.2 Long term trends, trend reversal and meteorological influences on the estimation of lockdown effects". *Meteorological influences are discussed in lines 4982-498.*

*This study follows up on our earlier work we published in Fan et al (2020) so it is logical that it is referenced. However, we have substantially reduced the text which summarized Fan et al. (see Introduction, line 76-94).*

I suggest to the co-author team to consider what the main take away message should be and re-write/re-present their work accordingly.

*R: At the end of the "Introduction", Lines 115-122, we have formulated 4 objectives and lines 123-127 describe how these are addressed. The "Discussion" sections follows the same structure. In the final section "Summary and conclusions" we start with a brief summary of work done and then structure Sect. 5 "Summary and conclusions" with bullets addressing these objectives. In particular we wrote:* "To answer these questions, satellite and ground-based data were analyzed for 11 regions in east China, leading to the following conclusions." *(Lines 605-607), and then listed these bullets (Lines 608-649).*

---

## Author Response (AR2)

**Fan et al., "Variability of NO$_2$ concentrations over China and effect on air quality derived from satellite and ground-based observations", submitted for publication in ACP. Revision 2.**

**Response to interactive comments to revision 1:**

We thank the Editor and Referees for their time to carefully consider our manuscript and provide comments. We have carefully considered the comments and revised the manuscript accordingly, in both rounds #1 and #2, and feel that the manuscript has been improved substantially. Our comments are provided below. The referee comments for round #2 are inserted in red, and our comments to each comment are written in black. In addition to responses to referee comments, some changes have been made in the revised text, and these are also added using "track changes".

**Referee #1**

I think the author seems to have misunderstood my previous comments. Original comments said "the content of this article hardly supports the title". Authors should supplement the content of the manuscript to echo the title, not reduce the content and change a title. Compared to the original manuscript, the current version is indeed largely up to the standard of publication, but still far from the ACP standard. The biggest shortcoming is that the current manuscript is not innovative enough. It is more of a continuation exercise than a new breakthrough. Unfortunately, I cannot recommend accepting this manuscript.

In the first revision we have responded to the comments of both reviewers and explained the revisions in our rebuttal. We have acknowledged that the previous "title did not fully cover the content". In such case, either the content could be changed, or the title. We decided to do the latter. In addition, as suggested by both Referees, also the content was reduced and structured.

In the comments to revision 1, Referee #1 mentions that "the current manuscript is not innovative enough" but does not explain what "enough" is and that "it is more of a continuation exercise than a new breakthrough". The referee has not mentioned this in the first review, and we strongly disagree with this comment. We have clearly listed our research questions and described in detail the approach taken to answer them, then discussed the results and provided conclusions to each research question (points 1-6 in Sect. 5). Indeed these may not be a breakthrough, but they are a solid piece of research, starting from earlier work and providing new results which are of interest for the wider AQ research community. We also discuss the implications of these new results. They are applicable, and addressing differences, over a large area with a variety of conditions (socio-economic and geographical and meteorological and climatological) and the results are also applicable over other areas than China. Hence, we do not understand these comments by Referee #1. In particular, we disagree that the manuscript does not meet "the ACP standard", and unfortunately the Referee has not indicated what this is either.

**Referee #2:**

The authors have made an excellent job both in addressing and replying both reviewers' comments as well as altering their manuscript to now better demonstrate the status of air quality over East China. I highly commend them on that. Having said that, I am still concerned about the presentation of their findings.

We thank referee #2 for the kind and constructive comments to further improve the manuscript. We feel that indeed the suggested changes have contributed to better presentation of the study and its results. Responses are provided below, after each comment. Line numbers refer to the tracked pdf version of the revised manuscript (file: Fan-etal-ACPD-Rev2_Final-tracked)

1. I would expect to first see the back-in-time discussion for the NO2 levels, i.e. Section 3.1.2, on OMI, and then what happened during 2020 and COVID, by TROPOMI, i.e. Section 3.1.1. It makes more of a natural flow of information this way, especially since you do not have back-in-time analysis for the AQI/surface concentrations.

Thank you for this comment. Indeed it makes sense to first go back in time and then concentrate on recent events. Also as regards the reduced concentrations in response to policy measures and the sudden change in response to COVID-19 measures. We have switched Sections 3.1.1 and 3.1.2 as suggested. For clarity, we have not tracked changes while switching. For further changes in these texts we did use "tracking": after switching, changes were needed in these 2 sections for a logical and chronological storyline. Also some text was added to the Introduction to further emphasize this (lines 73-80). Figure numbers have been changed and we have carefully checked that also in the rest of the text these changes were made.

2. Figure 4 shows OMI NO2 up to and including months of year 2020, these should be excluded since you do not plan to discuss them. Overall the quality of the graph is poor, make sure all labels increase in size, are made bold, and so on.

Thank you for this comment. The quality of Figure 4 (Figure 2 in the revised version) has been improved and the time series now ends in December 2019.

Furthermore, when a trend is calculated from such a seasonally-varying species as is NO2, the time series has to be de-seasonalised beforehand. This is not explicitly mentioned in the text. I strongly suggest that you first de-seasonalise and then provide trends in the relevant Table.

Thank you for this comment. Instead of de-seasonalizing we had used annual averages. In the current version (revision#2), we have de-seasonalized the monthly time series using a centered moving average with a period of 12 months, before calculating annual mean values (now mentioned on line 294-297). We decided for the annual mean values to reduce effects of variations due to other influences than the seasonal effects. Time series plots of the annual mean $NO_2$ TVCDs, without the trend lines, are presented in Figure A1, to better illustrate our point that there are differences between the time series in the north and in the south. Then the trends were re-calculated for certain intervals (Table 2, and described in the text) added to the annual mean time series (Figure 3 in the revised version) and the new trend values are presented in table 2. The features in the time series and trends were similar to those in the previous version, with some small changes.

3. Since in the rest of the text you show weekly timeseries for the AQI/surface concentrations for the COVID weeks, I strongly recommend you also show, in the TROPOMI section, similar timeseries. If not, again your results appear to be "un-connected' i.e. you first have one satellite giving you the 10 year NO2 trend, then another giving you what went on during COVID for the 2020 weeks, then you have surface concentrations of NO2 and other species, plus AQI, only for the COVID 2020 weeks. I am not suggesting that you compare TROPOMI NO2 to the surface concentrations, this task is a paper on its own. You should however show if similar patterns appear. These patterns cannot be seen from Fig. 3 & Fig A1 which are demonstrational only.

Thank you for this suggestion, it was a very good idea and Section 3.3 has been added with a Figure 9, showing the weekly averaged $NO_2$ concentrations as time series for week -3 to week 20, both for TROPOMI TVCDs (averaged over each region) and the ground-based data (replotted from Figures 6-8 and 10. We do this comparison only for the selected regions in each of the groups 1-3 , and Beijing, for clarity and easy comparison. Adding more regions made Figure 9 too full to be clear. The comparison is only qualitative since a quantitative comparison would require the consideration of meteorological condition (through a model). Nevertheless, the qualitative comparison shows some interesting differences and similarities, as described in Section 3.3. We have also added some text in the Abstract (line 37-39), Introduction (lines 150-151) and Summary and Conclusions (lines 827-829) to this effect.

Overall, from the time series provided, I note that the AQI is mostly influenced by PM2.5, rather than the variation on NO2.

This indeed is true, and was mentioned in Sect. 4.3 lines 735-736 ("However, the latter were not reflected in the AQI, which followed the variations in $PM_{2.5}$ but remained low when $O_3$ concentrations were high"). We realized that this comment was somewhat hidden in the text and therefore we have added some words reflecting this at other places: Abstract (line 37-39), Discussion (lines 770-771) and Summary and Conclusions (Lines 873-874).

Technical: throughout the text there are references to a Figure/Table in the supplement, I assume this was a typo error

Thank you for this observation: indeed it was a typo remaining from our initial editing, before we had read in the Instruction that it should be called Appendix. We have checked the manuscript and corrected these typos.

Figure A1 should be made with a colour bar the reflects differences, i.e. going from blue to white - which means 0- and then to red

This is not correct, we subtracted the TVCD map for each week from that for week 0, and hence green colours indicate a decrease with respect to week 0, red colours an increase, but it is still a TVCD. This was also explained in the text in Sect. 2.1 (lines 165-166), where we describe Fig. 1. To make it clear, we have added this information to the caption of Figure A2.

Figure A2 is of low quality and should be improved.

Thank you for this comment. We have replotted Figure A2 (Figure A1 in the revised version) similar to Figure 5 (Figure 5 in the revised version).